# The nucleocytoplasmic translocation of HINT1 regulates the maturation of cell density

Xiaofang Zhang ⬡, Fumihiko Nakamura ⬡

In normal epithelial cells on tissue culture dishes, contact inhibition typically progresses with a reduction in cell size after cell–cell contact. This transition involves actin cytoskeleton reorganization from stress fibers (SFs) to a cortical network, stabilizing cell shape and strengthening connections. However, the regulatory signaling pathways remain unclear. We identified histidine triad nucleotide-binding protein 1 (HINT1), also known as protein kinase C inhibitor 1 (PKCI-1), as a regulator for monolayer maturation. At low density, HINT1 localizes in the nucleus and binds to open chromatin. As density increases, exportin 1 drives HINT1 translocation to the cytoplasm. Forced cytoplasmic localization of HINT1 reduces phosphorylation of myristoylated alanine-rich C kinase substrate (MARCKS) at Ser167/170, sites specifically targeted by PKC and involved in regulating SF formation. MARCKS phosphorylation also decreases naturally at high density. Although cells can form a monolayer without HINT1, its presence is required for full cell confinement and mature monolayer formation. Thus, HINT1 plays a dual role: acting as a nuclear transcriptional coregulator at low density and acting as a cytoplasmic SF modulator at high density.

## Introduction

Early unicellular organisms evolved to proliferate indefinitely under favorable environmental conditions, such as sufficient nutrient availability. In contrast, multicellular organisms developed regulatory mechanisms that suppress unchecked cell division, thus enabling the formation of distinct body architectures. Cancer arises when these regulatory mechanisms fail, resulting in the uncontrolled expression of intrinsic proliferative capacities. Such inhibitory mechanisms are known as contact inhibition of proliferation (CIP) (Stramer & Mayor, 2017; Nakamura, 2024; Carlantoni et al, 2025). In general, CIP in normal epithelial cells cultured on tissue culture dishes progresses through at least three distinct stages: (i) a phase of increasing cell density during which cell motility declines (contact inhibition of locomotion, CIL) but mitosis persists; (ii) a rapid transition to an epithelial morphology; and (iii) continued cell division that progressively reduces cell size (cell density maturation) and slows the mitotic rate. Ultimately, mitosis ceases once the cell area drops below a critical threshold (Puliafito et al, 2012). These observations suggest that CIP arises from mechanical interactions and constraints rather than simple cell–cell contact. However, the molecular mechanisms that regulate these processes, particularly the final stage, remain largely unexplored.

During these transitions, the actin cytoskeleton undergoes remodeling. When tissue culture cells are grown at low densities, these cells migrate and frequently exhibit polarized stress fibers (SFs). As cell density increases during high-density culture, SFs tend to disappear, and filamentous actin (F-actin) is redistributed to the cell cortex, resembling its in situ distribution (Rubin et al, 1978; Toh et al, 1979; Hormia et al, 1985; Bereiter-Hahn & Kajstura, 1988; Kajstura & Bereiter-Hahn, 1989; Jou & Nelson, 1998; Li et al, 2019). These arrangements suppress cell spreading and movement and support maintenance of cell shape and reinforcement with neighboring cells.

During CIL and CIP, various molecules, such as cadherins, focal adhesion kinase, and small GTPases, remodel the cytoskeleton to promote monolayer formation (Knight et al, 1995; Koyama et al, 1996; Nobes et al, 1998; Wassler & Shur, 2000; Stramer & Mayor, 2017; Nakamura, 2024). Our findings reveal that histidine triad nucleotide-binding protein 1 (HINT1), also known as protein kinase C inhibitor 1 (PKCI-1) (McDonald & Walsh, 1985; Robinson & Aitken, 1994), plays a role in further maturing cells within the monolayer, leading to smaller cell shapes. HINT1, a tumor suppressor, is involved in various functions such as cell signaling and gene expression (Dillenburg et al, 2023). For example, HINT1 acts as a transcriptional coregulator, influencing gene expression by interacting with various transcription factors such as MITF (microphthalmia-associated transcription factor) and β-catenin to regulate cell proliferation, apoptosis, and differentiation (Genovese et al, 2012). Here, we demonstrate that at low cell density, HINT1 primarily localizes in the nucleus, where it binds to chromatin. As cell density increases, HINT1 translocates to the cytoplasm—independent of direct cell–cell contact—where it likely contributes to stress fiber (SF) disassembly by inhibiting phosphorylation of MARCKS at Ser167/170, sites specifically targeted

School of Pharmaceutical Science and Technology, Faculty of Medicine, Tianjin University, Tianjin, China

Correspondence: fnakamura@tju.edu.cn

by PKC. Supporting this model, forced cytoplasmic localization of HINT1 via mutation of its NLS leads to reduced SFs and promotes a rounded cell morphology over an extended one. Similarly, pharmacological inhibition of exportin 1, which traps HINT1 in the cytoplasm, results in comparable morphological changes. Furthermore, HINT1 loss increases cell cross-sectional area at high density, consistent with hypertrophy seen in cardiomyocytes of HINT1 knockout mice and in patients with low HINT1 expression, though the mechanism was previously unclear. These findings identify HINT1 as a key regulator that dynamically shuttles between the nucleus and the cytoplasm in response to cell density. Our data suggest that HINT1 promotes mature monolayer formation at high density by remodeling the actin cytoskeleton through inhibition of MARCKS phosphorylation, potentially explaining previously observed three-phase progression of contact inhibition.

## Results

### HINT1 translocates between the nucleus and the cytoplasm in a cell density–dependent manner

We recently developed a DSP-MNase-proteogenomics method to identify proteins that bind to open chromatin (Li et al, 2023). HINT1 was identified as a potential chromatin-binding protein in low-density human skeletal muscle (hsSKM) cells, with reduced binding observed in high-density cells (Table S1 in Li et al [2023]). In this study, we confirmed whether HINT1 can shuttle between the nucleus and the cytoplasm in a cell density–dependent manner. To address this, cells were seeded across a wide range of densities to effectively assess the impact of cell density on the subcellular localization of HINT1 and to identify overall trends, while acknowledging the challenges of defining precise confluency. Although many cell types typically reach full confluency at around $1–1.5 × 10^5$ cells/cm$^2$ on tissue culture dishes, 100% confluency cannot be strictly determined by cell number alone, as it varies depending on the cell type. Some cells continue to proliferate and adopt a more compact morphology after initial contact, making it difficult to establish a consistent threshold. In our preliminary experiments (Fig S1A), we observed that seeding HEK293A cells at $0.7 × 10^5$ cells/cm$^2$ resulted in nearly all cells contacting one another after 24 h, which we defined as 100% confluency. At this density, cells form intercellular adhesions, as indicated by β-catenin staining; however, the cross-sectional cell area continues to become smaller even after cell adhesion is established (Fig S1B). In addition, the extent of further crowding beyond initial contact was difficult to quantify, and in some cases, small gaps remained. Therefore, a seeding density of 200% refers to twice the number of cells required to reach 100% confluency in our study. Because not all seeded cells attach, values like 400% do not represent actual confluency but rather the relative number of cells initially plated. Unless otherwise specified, we used 20% and 400% seeding densities to represent low and high cell densities, respectively. We chose 400% as the high-density benchmark because the cells reached their minimum cross-sectional area at densities exceeding this value (Fig S1B).

First, we expressed exogenous HINT1 with a hemagglutinin (HA) tag attached to the C terminus in HEK293A cells at both low and high densities (Fig 1A and B). To assess whether the molecular mechanism we discovered shows cell-type specificity, we attempted to use various types of cultured cells. However, because of experimental limitations such as transfection difficulties, we were constrained to using specific cell types. Accordingly, we have clearly specified which cell type was used in each experiment. Consistent with the proteomics data, HINT1-HA was primarily localized in the nucleus of the transfected HEK293A cells at low density, whereas it was found in the cytoplasm at high density. To visualize the localization of endogenous HINT1 in various tissue culture cells, we used a commercially available specific antibody (Fig 1C). Cell density did not affect the expression level of HINT1 or induce fragmentation. Consistent with the results for exogenously expressed HINT1-HA, endogenous HINT1 was also primarily localized in the nucleus of HEK293A, hsSKM, and MEF cells at low density, with slight variations depending on the cell type (Figs 1D and E and S2).

Cell–cell contact can induce CIP through various transmembrane proteins (Stramer & Mayor, 2017; Nakamura, 2024). However, simple cell confluency is not sufficient to trigger Yes-associated protein (YAP) translocation to the cytoplasm; much higher cell density is required for this process to occur (Aragona et al, 2013). Although HINT1 remains in the nucleus when cells are in partial contact (e.g., 70% confluency in Fig S3A), near full confluency (e.g., 100%) is sufficient to initiate its translocation to the cytoplasm in hsSKM cells. As cell density increases, more HINT1 translocates to the cytoplasm (Fig S3B). This coincided with a density-dependent reduction in cross-sectional cell area (Fig S1B), as HINT1's cytoplasmic localization maximized at 300–400% density (Fig S3B). In addition to these untransformed cells, we also tested localization of HINT1 in transformed HeLa cells at various densities (Fig S4). At high density, HINT1 predominantly relocated to the cytoplasm in HEK293A, hsSKM, and MEF cells, whereas in HeLa cells, nuclear localization remained more prominent compared with the other cell types (Fig S4A and B).

### HINT1 localization is not responsive to mechanical cues

Many nucleocytoplasmic molecules, such as YAP and core-binding factor subunit beta (CBFB), display both density-dependent localization and mechanosensitivity, which is influenced by factors such as actin–myosin contraction and substrate stiffness (Dupont et al, 2011; Li et al, 2024; Nakamura, 2024). To determine whether HINT1 shares this sensitivity to internal mechanical stress, we treated cells with latrunculin B (an actin polymerization inhibitor) and blebbistatin (a myosin II inhibitor). Unexpectedly, HINT1 translocation was unaffected by these treatments in HEK293A, hsSKM, and MEF cells at both low density (Figs 2A and B and S5A–C) and high density (Fig S6A and B). In contrast, as expected, YAP translocated to the cytoplasm at low density after these drug treatments (Fig S7A and B), consistent with previous findings (Dupont et al, 2011). We also assessed HINT1 translocation in cells cultured on a soft substrate, but even on a 0.2 kPa substrate, HINT1 did not translocate to the cytoplasm in hsSKM cells (Fig 2C and D).

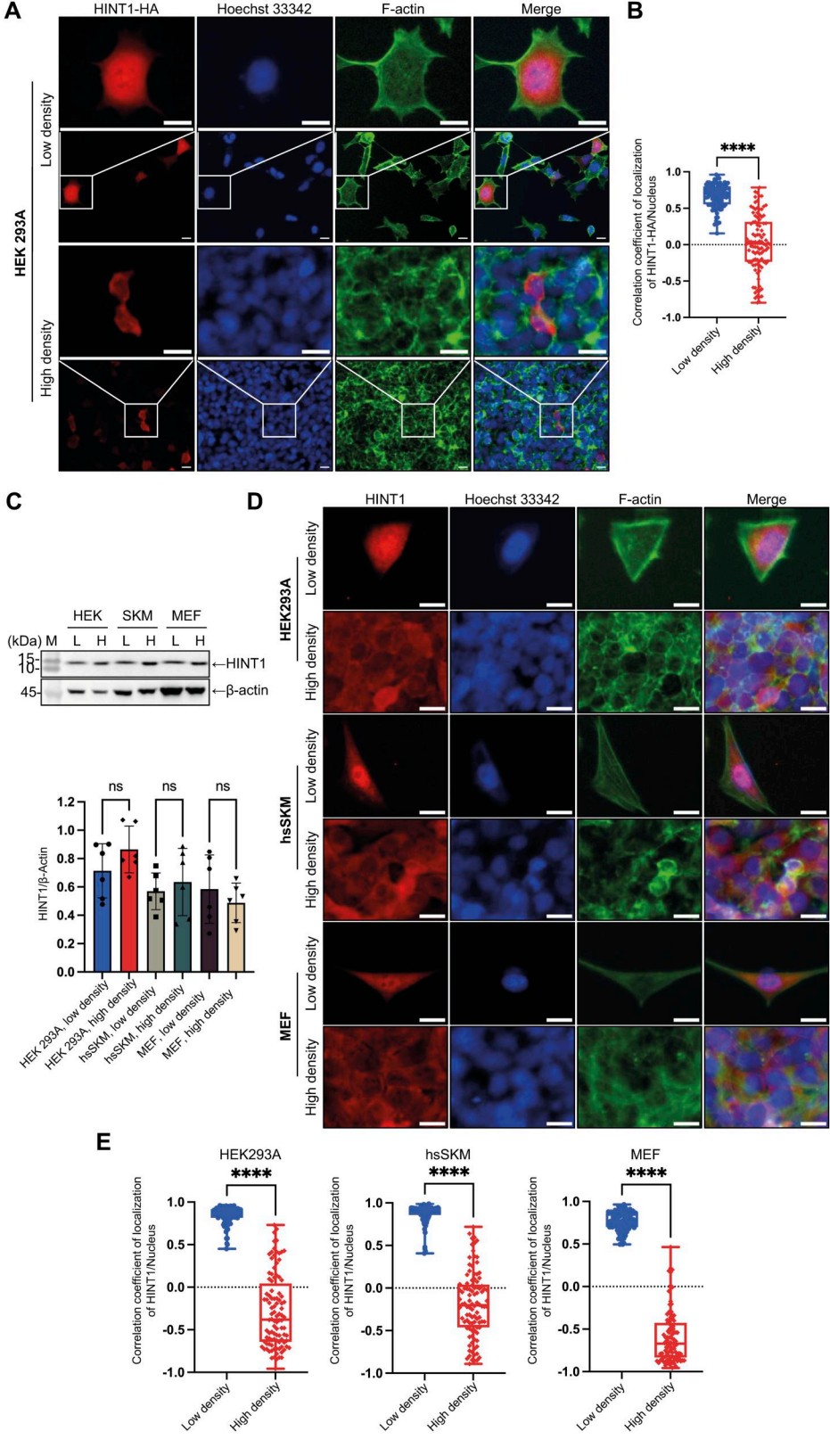

**Figure 1. HINT1 translocates between the nucleus and the cytoplasm in a cell density–dependent manner.**
**(A)** Exogenously expressed HINT1-HA in HEK293A cells at low density mainly localizes in the nucleus, whereas it translocates to the cytoplasm at high density. HINT1-HA was stained with rabbit anti-HA antibody followed by anti-rabbit IgG Alexa Fluor Plus 594 antibody (red). Nuclear DNA was stained with Hoechst 33342 (blue). Actin filaments were stained with phalloidin Alexa Fluor 488 (green). Scale: 20 $\mu$m. **(B)** Correlation coefficient of HINT1-HA staining and Hoechst nuclear staining was calculated and plotted (n = 100). **(C)** Expression of HINT1 in various tissue culture cells detected by Western blotting using the specific antibody against HINT1. **(D)** Localization of endogenous HINT1 expressed in various tissue culture cells at low and high densities. HINT1 was stained with rabbit anti-HINT1 antibody followed by anti-rabbit IgG Alexa Fluor Plus 594 antibody. Nuclear DNA was stained with Hoechst 33342. Actin filaments were stained with phalloidin Alexa Fluor 488. Scale: 20 $\mu$m. **(E)** Correlation coefficient of HINT1 staining and Hoechst 33342 staining was calculated and plotted (n = 100).
Source data are available for this figure.

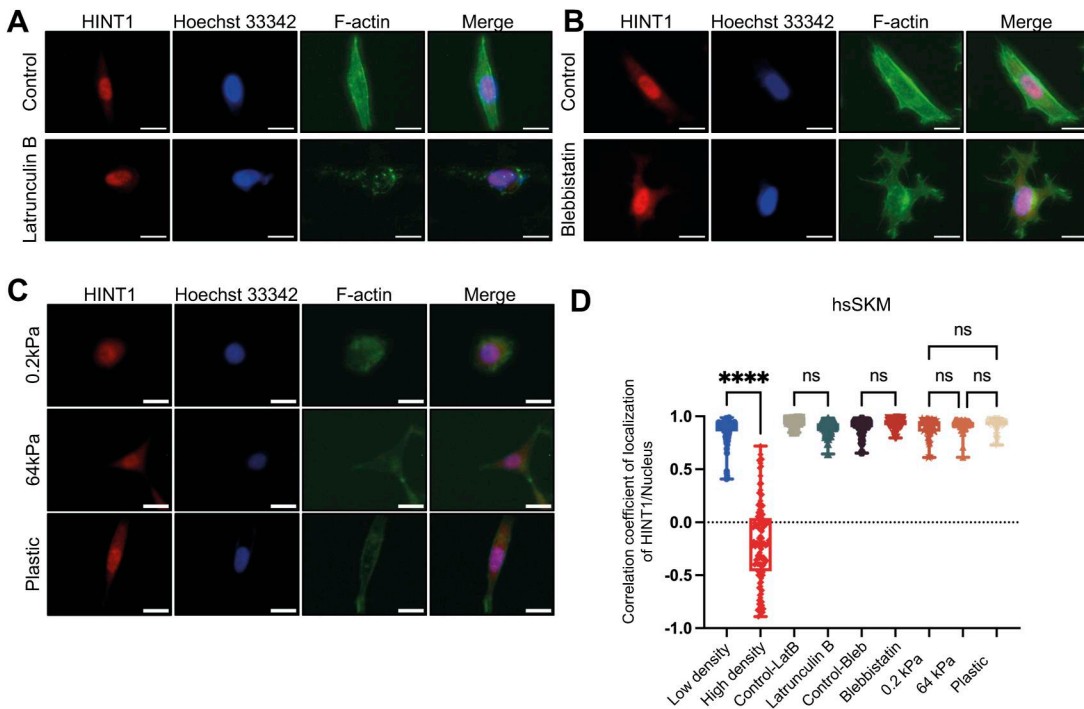

**Figure 2. HINT1 localization is not responsive to depolymerization of actin and mechanical cues.**
**(A, B, C)** Cells were stained as shown in Fig 1D. Scale: 20 µm. **(A)** hsSKM cells were treated with or without 5 µM of latrunculin B for 60 min. **(B)** hsSKM cells were treated with or without 50 µM of blebbistatin for 180 min. **(C)** hsSKM cells were cultured on soft (0.2 kPa) and stiff substrate (64 kPa and plastic). **(D)** Correlation coefficient of HINT1 staining and Hoechst 33342 staining was calculated and plotted (in the 0.2, 64 kPa, and plastic groups, n = 50, whereas in the other group, n = 100). Related to Fig S5.

## Loss of HINT1 delays cell density–dependent cell death but not cell migration

To explore the function of HINT1, we generated HINT1 knockout (KO) HEK293A cells using the CRISPR-Cas9 system. Western blotting and immunofluorescence microscopy verified that three cloned KO cells lack HINT1 expression (Fig 3A and B). Unexpectedly, no significant morphological changes were observed following the knockout (Fig 3B). Consequently, in the subsequent related experiments, only the KO-1-E9 cell line was used, and all HINT1 KO cells refer to the KO-1-E9 cell line, unless otherwise specified. Next, we examined whether the HINT1 knockout affects cell migration (Fig 3C–E). Once again, no significant changes were observed. When each cell was cultured with daily fresh medium, no significant difference in proliferation was observed for the first 10 d. However, WT cells died after 6 d of reaching a peak. Interestingly, KO cells survived longer than WT cells (Fig 3F).

Finally, we analyzed differentially expressed genes (DEGs) between WT and HINT1-KO cells using RNA-seq analysis (Table S1). The results showed that HINT1 KO led to both the up-regulation and down-regulation of genes associated with morphogenesis and development (Figs 3G and H and S8A–G). The volcano plot highlighted significant up- and down-regulated genes between WT and HINT1-KO cells, with the up-regulated genes being primarily involved in neural development (Fig 3I), potentially linked to *HINT1* gene polymorphisms in human

behaviors and neuropathy (Morel et al, 2022; Suchanecka et al, 2024). However, further analysis is necessary to confirm the functional relevance of these gene expression changes in the context of HINT1-related neurological phenotypes. For example, the expression of different collagen genes changed in HINT1-KO cells; however, because some of these genes are up-regulated, whereas others are down-regulated under the KO condition, we assume that collagen is not a definitive factor in cell density regulation, assuming these collagens have similar roles in cell adhesion.

## HINT1 regulates cell cross-sectional area at high density

Although no significant differences in cell morphology, proliferation, or migration were observed between WT and HINT1 KO cells at low density, KO cells survived longer than WT cells at high density, suggesting that HINT1 functions under high-density conditions. First, we stained actin filaments in WT and KO cells at various densities (Fig S9). Although no significant differences in actin distribution were observed between WT and KO cells, we noticed that the cell area of KO cells appeared larger than that of WT cells. To confirm this observation, cell boundaries were demarcated using a fluorogenic membrane dye to quantitatively assess cell cross-sectional area (Fig 4A and B). Consistent with the actin staining images, cross-sectional area of KO cells at high density is larger than that of WT cells, whereas at low density, such difference was not observed.

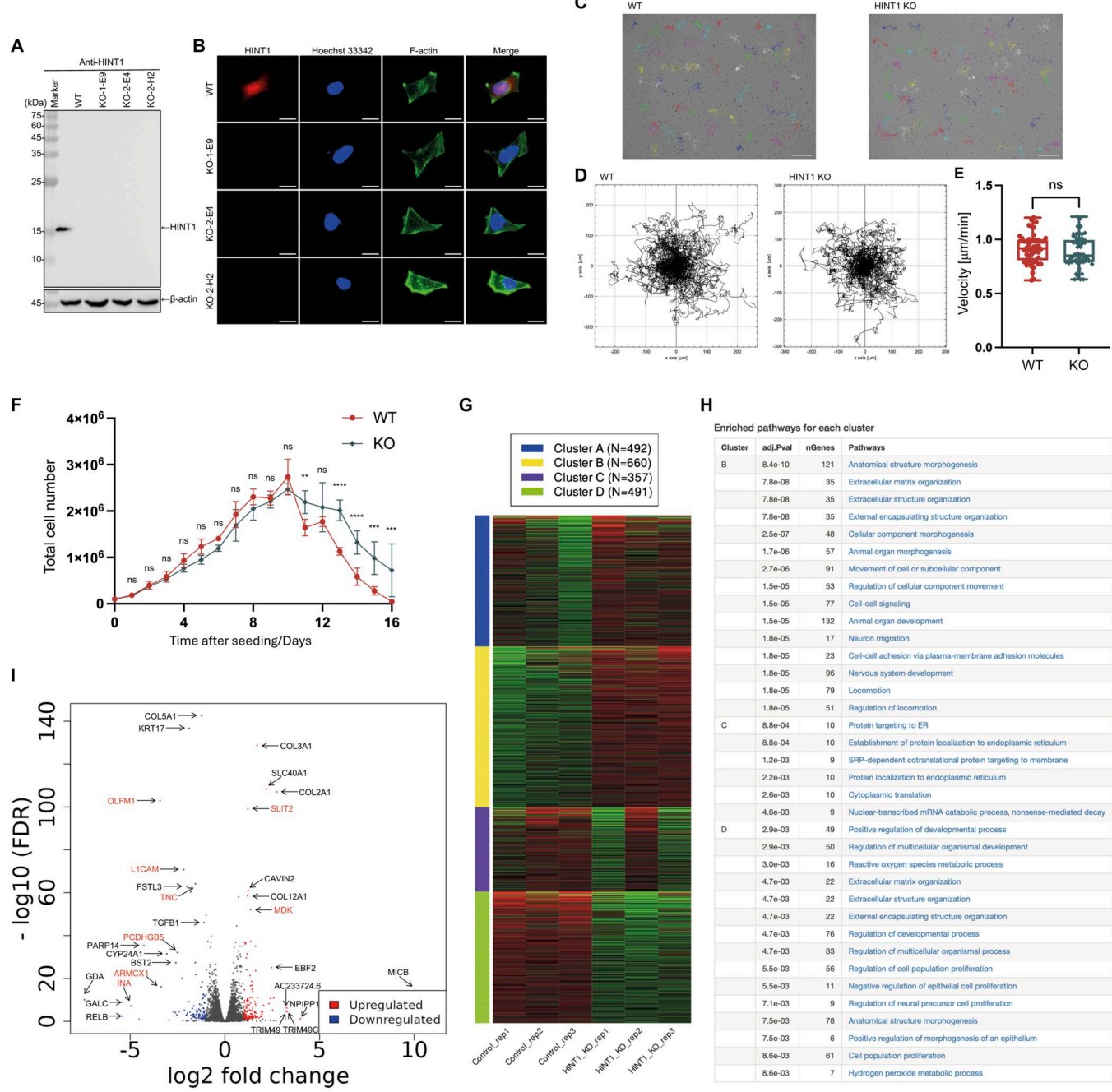

**Figure 3. Loss of HINT1 delay cell density–dependent cell death.**
**(A, B)** Western blotting and immunofluorescence microscopy confirm loss of HINT1 in three independent HINT1-KO cell lines. The cells were stained as shown in Fig 1D. Scale: 20 μm. **(C, D, E)** Images of cells were captured at 1 frame/10 min for 20 h by time-lapse microscopy and tracked the migratory paths of cells. Scale: 200 μm. **(D)** Plots show an example tracking 50 cells from one of three independent experiments. **(E)** Mean migration speed (μm/min) of cells. Values represent the migration speed of each cell as a dot. Statistically not significant (ns) by two-tailed unpaired Student's test. **(F)** Equal number of WT and HINT1-KO HEK293A cells were cultured in a 24-well tissue culture plate, and the medium was replaced every day. At the indicated time after seeding, cell numbers were counted and plotted. Data represent an average of three independent experiments. Statistical analysis was performed by two-way ANOVA. ns, not significant; *$P \leqq 0.05$; **$P \leqq 0.01$; ***$P \leqq 0.001$; ****$P \leqq 0.0001$. **(G, H)** DEGs between WT and HINT1-KO cells. K-means clustering (the red and blue represent significantly up- and down-regulated genes between WT and HINT1-KO cells, respectively) and enrichment analysis. Cluster A is not shown, as it exhibits substantial variation within the same cell group (WT versus KO). **(I)** Volcano plot of false discovery rate (−$\log_{10}$ FDR) against $\log_2$ (fold change). The red and blue dots represent significantly up- and down-regulated genes between WT and HINT1-KO cells, respectively. The genes indicated in red are involved in neural development.

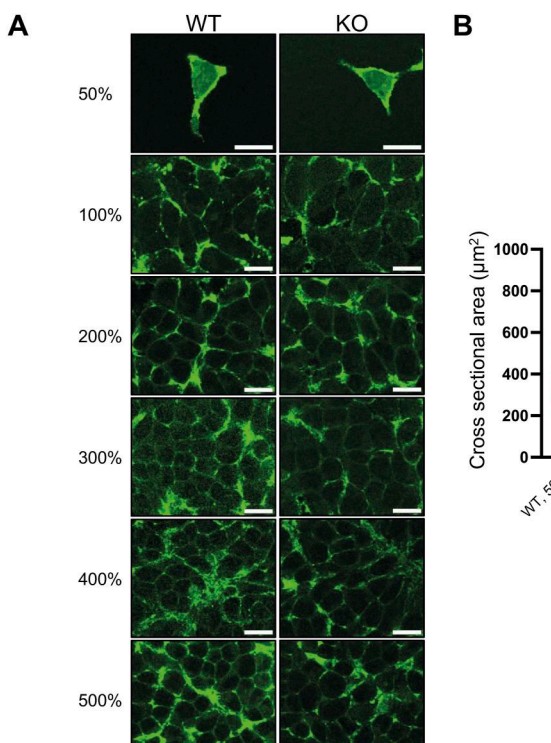

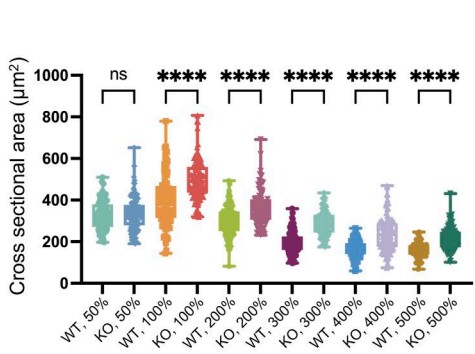

**Figure 4. HINT1 KO enlarges the cell cross-sectional area at high density.**
**(A)** WT and HINT1-KO HEK293A cells were stained with CellBrite Fix 488 fluorogenic membrane dye to demarcate the cell boundaries at various cell densities. Scale: 20 $\mu m$. **(B)** Values represent the cross-sectional area of each cell as a dot (n = 100). The percentages are values used for convenience when seeding cells and do not represent the actual cell density. However, for 100%, the cell count was measured and cultured in such a way that the cells would reach confluency by the time of fixation.

## Forced cytoplasmic expression of HINT1 disrupts actin SF and promotes a rounded, confined cell morphology

To investigate molecular mechanism of how translocation of HINT1 to the cytoplasm regulates cell morphology at high density, we engineered HINT1 that is expected to be consti-tutively expressed in the nucleus or cytoplasm. Although at-tachment of the well-known NLS derived from SV40 large T antigen (NLS, MPKKKRKV–, referred to as the added NLS or aNLS) to the N terminus of HINT1 promotes translocation of HINT1 to the nucleus in low-density cells, the engineered HINT1 still stay in the cytoplasm at high density in HINT1-KO HEK293A cells (Figs S10A–C and S11A and B) (Kalderon et al, 1984). A mutation in the aNLS sequence (MPK<u>T</u>KRKV–), which is expected to disrupt nuclear localization (Kalderon et al, 1984), only slightly reduced the aNLS's effect on nuclear transloca-tion at low density, without significantly altering the overall intracellular distribution of HINT1 in HINT1-KO HEK293A cells (Fig S11A and B). This suggests that the HINT1 protein contains an endogenous nuclear localization signal (eNLS). As expected, point mutation of the predicted eNLS (Fig S10D) in the HINT1 molecule disrupted the nuclear localization in HINT1-KO HEK293A cells (Fig 5A and B). Furthermore, in MEF cells, the exogenously expressed HINT1-HA with the R24A/R25A mutation impaired actin SF (Fig S12) and significantly altered cell mor-phology, causing the cells to become smaller, rounder, and more compact (Fig 5C and D).

## Translocation of HINT1 to the cytoplasm is regulated by exportin 1 downstream of high cell density

Point mutation of intramolecular eNLS of HINT1 disrupted its nuclear translocation, suggesting that active transport is involved in the nucleocytoplasmic shuttling of HINT1. First, we tested two available importin inhibitors—ivermectin (an importin $\alpha/\beta 1$ complex inhibi-tor) and importazole (an importin $\beta$ inhibitor) (Jans et al, 2019). However, neither showed a significant effect on the localization of HINT1 in HEK293A, hsSKM, and MEF cells at low density (Fig S13A and B). Next, the cells were treated with leptomycin B, an exportin 1/ CRM1 inhibitor, at both low and high cell densities. Leptomycin B did not affect the nuclear localization of HINT1 at low density but inhibited its translocation to the cytoplasm at high density in HEK293A, hsSKM, and MEF cells (Figs 6A and B and S14A–D). At high cell density, the inhibition of exportin 1 led to an increase in cell cross-sectional area in WT cells, but this effect was not observed in HINT1 KO cells (Fig 6C and D).

## Translocation of HINT1 to the cytoplasm inhibits PKC

Although increased PKC activity has been observed in HINT1-deficient mice (Garzon-Nino et al, 2017), the role of HINT1 translocation in regulating PKC activity has not yet been explored. We hypothesized that PKC activity is elevated at low cell density because of HINT1's localization in the nucleus, whereas at high density, PKC activity decreases as translocated

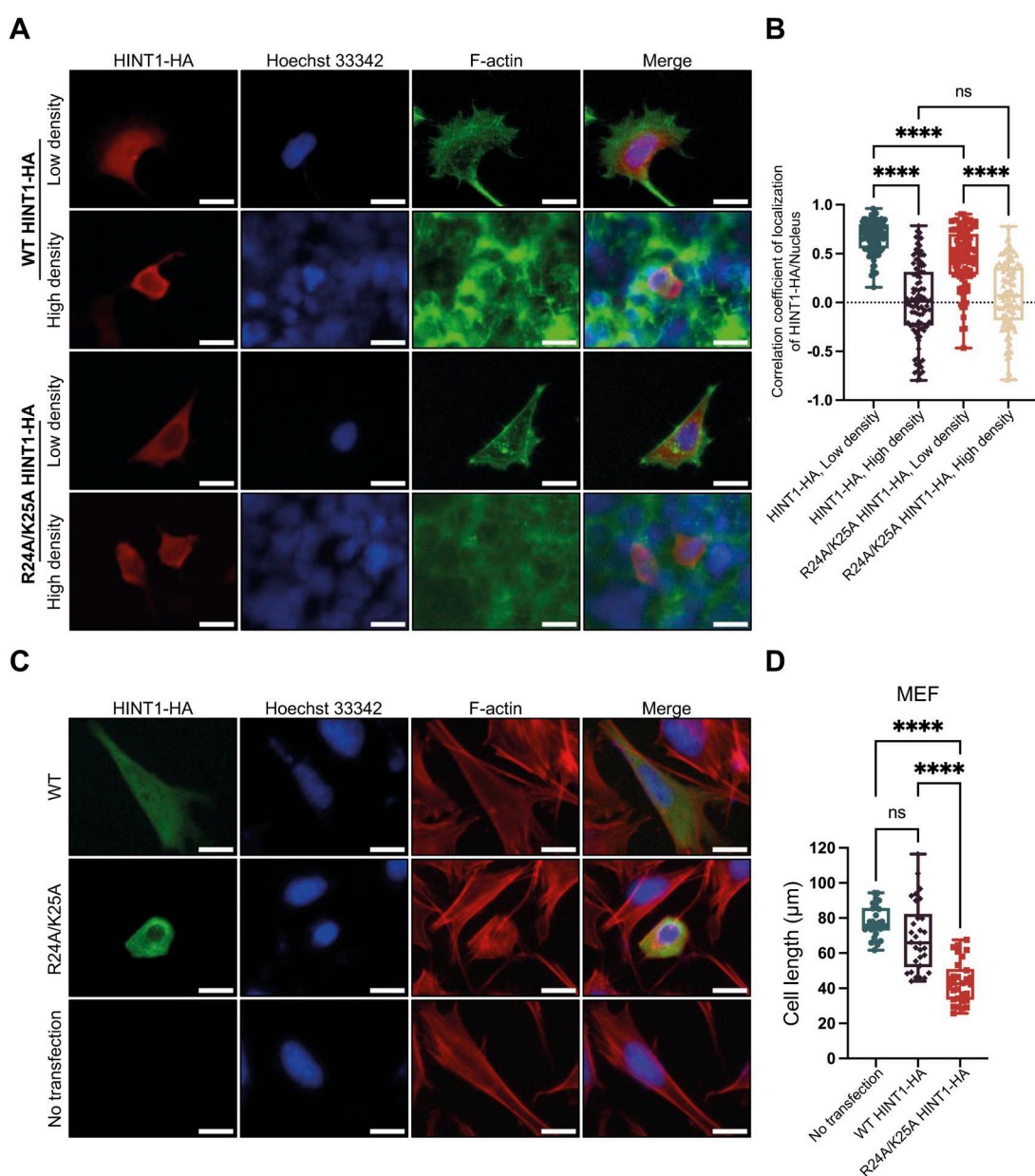

**Figure 5. Forced expression of HINT1 in the cytoplasm disrupts elongation of cells.**
**(A, B)** Exogenously expressed HINT1-HA in HINT1-KO HEK293A cells at low density primarily localizes in the nucleus, whereas the point mutation in the intramolecular eNLS (R24/K25A) promotes its cytoplasmic localization. The cells were stained as shown in Fig 1A, and the correlation coefficient between HINT1-HA staining and Hoechst nuclear staining was calculated and plotted. Scale: 20 $\mu$m. **(C, D)** MEF cells were transfected with WT and mutant R24A/K25A HINT1-HA at low density and stained with rabbit anti-HA antibody followed by anti-rabbit IgG Alexa Fluor Plus 488 antibody (green). Nuclear DNA was stained with Hoechst 33342 (blue). Actin filaments were stained with phalloidin Alexa Fluor 568 (red). Notably, the mutant HINT1 localized in the cytoplasm and disrupted the elongated shape of the cells, whereas the WT HINT1 did not affect cell morphology. The length of the major axis of each cell was measured and plotted. Each dot represents the length of an individual cell (n = 30). Scale: 20 $\mu$m.

HINT1 inhibits PKC in the cytoplasm. Given that myristoylated alanine-rich C kinase substrate (MARCKS) is a major substrate of PKC, and that Ser167 and Ser170 are the only phosphorylation sites on MARCKS known to be specifically targeted by PKC to date (Hornbeck et al, 2004) (https://www.phosphosite.org/homeAction), we used a site-specific anti-phospho antibody to evaluate MARCKS phosphorylation at Ser167/170. In addition,

PKC-dependent phosphorylation of MARCKS plays a crucial role in cytoplasmic actin remodeling (Hartwig et al, 1992; Calabrese & Halpain, 2005). We first measured the phosphorylation levels of MARCKS in various cell lines at low and high densities using a phospho-MARCKS (Ser167/170) polyclonal antibody (Graff et al, 1989). As anticipated, the phosphorylation levels of MARCKS were significantly higher in low-density cells compared with

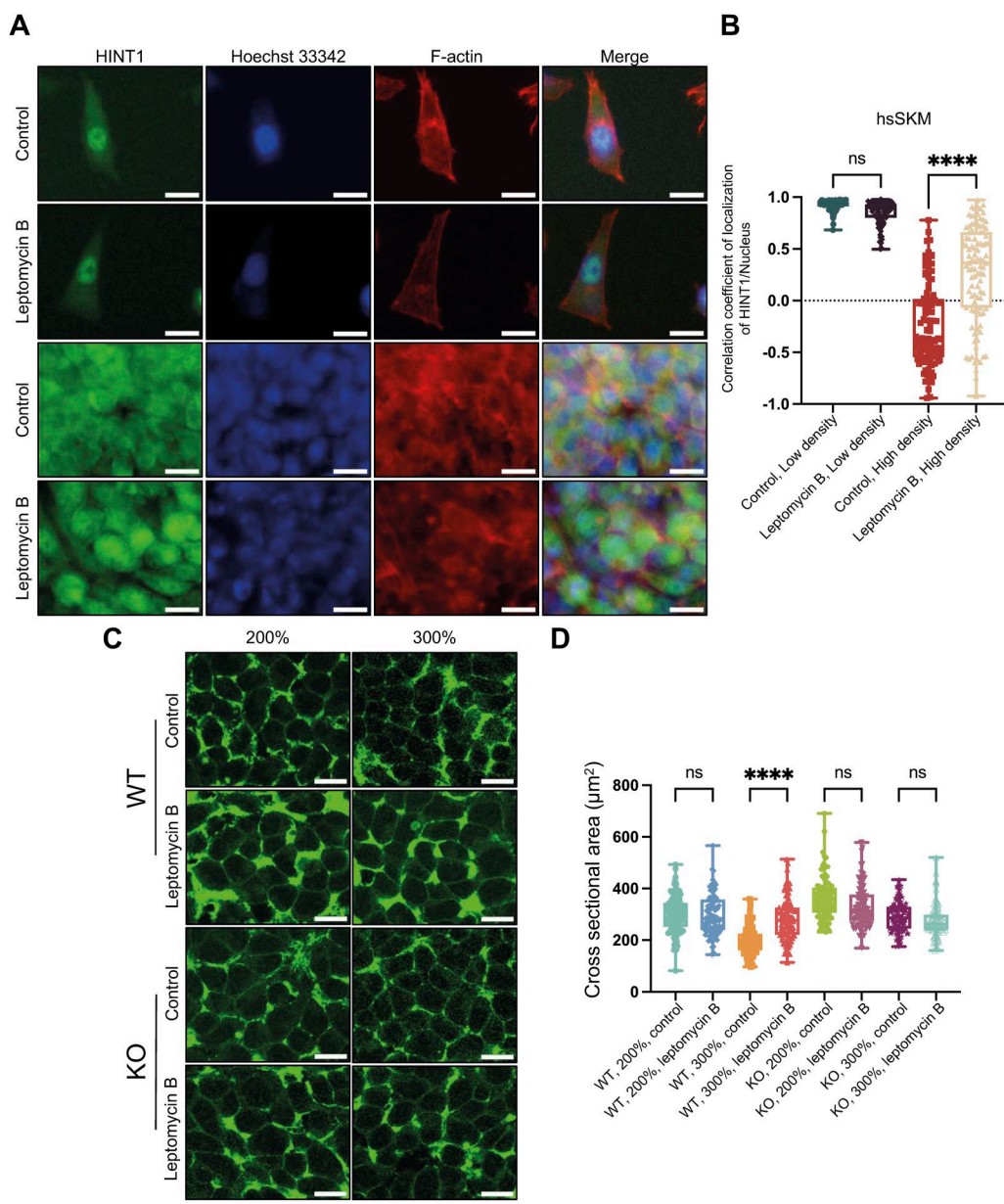

**Figure 6. Inhibition of exportin 1 by leptomycin B blocks the translocation of HINT1 to the cytoplasm at high cell density and increases the cell cross-sectional area.**
**(A)** hsSKM cells were treated with or without 10 nM of leptomycin B for 24 h at both low and high densities, then stained with rabbit anti-HINT1 antibody followed by anti-rabbit IgG Alexa Fluor Plus 488 antibody. Nuclear DNA was stained with Hoechst 33342. Actin filaments were stained with phalloidin Alexa Fluor 568. Scale: 20 μm. **(B)** Correlation coefficient of HINT1 staining and Hoechst nuclear staining was calculated and plotted. Similar results were obtained with HEK293A and MEF cells (Fig S14). **(C)** WT and HINT1-KO HEK293A cells were treated with or without 10 nM of leptomycin B for 24 h at high densities and stained with fluorogenic membrane dye to demarcate the cell boundaries. Scale: 20 μm. **(D)** Values represent the cross-sectional area of each cell as a dot (n = 100). The percentages are values used for convenience when seeding cells and do not represent the actual cell density. However, for 100%, the cell count was measured and cultured in such a way that the cells would reach confluency by the time of adding leptomycin B.

high-density cells in HEK293A, hsSKM, and MEF cells (Fig 7A). Next, we tested whether forced expression of HINT1 in the cytoplasm of low-density cells would also inhibit MARCKS phosphorylation in HEK293A cells. Compared with nontransfected cells, the overexpression of WT HINT1-HA led to a decrease in MARCKS phosphorylation, likely because of the presence of some HINT1 in the cytoplasm (Fig 7B). Supporting our hypothesis, the mutant form of HINT1, which predominantly localizes in the cytoplasm, resulted in an even greater reduction in MARCKS phosphorylation. Furthermore, we compared MARCKS phosphorylation levels between WT and HINT1-KO cells (Fig 7C). As expected, phosphorylation was significantly elevated in the KO cells compared with WT, further supporting our hypothesis.

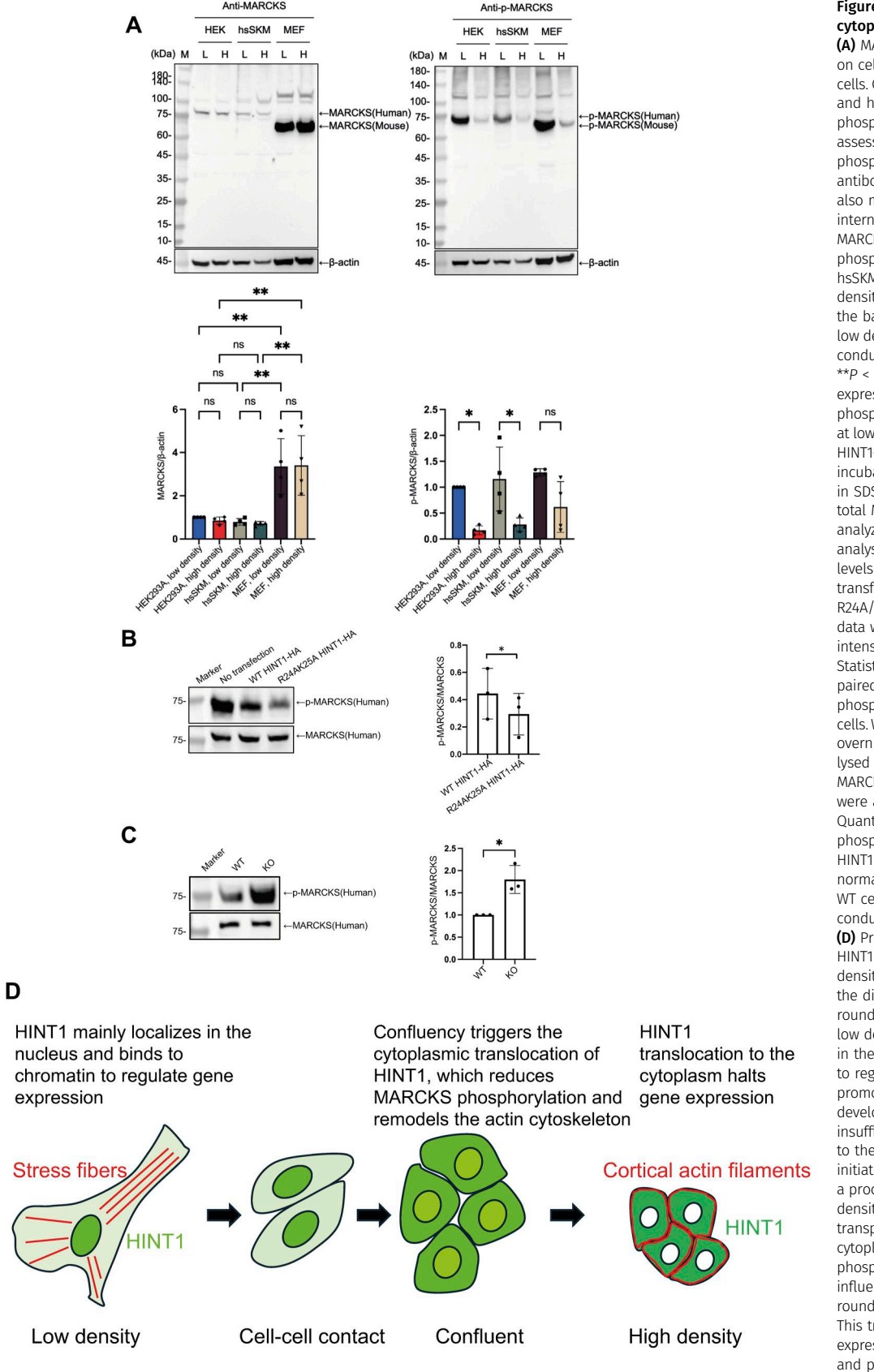

**Figure 7. Translocation of HINT1 to the cytoplasm inhibits PKC activity.**
**(A)** MARCKS phosphorylation is dependent on cell density in HEK293A, hsSKM, and MEF cells. Cells were cultured overnight at low and high densities, and the phosphorylation levels of MARCKS were assessed by Western blotting using a phospho-MARCKS (Ser167/170) polyclonal antibody. Total MARCKS and actin levels were also measured via Western blotting as internal controls. Quantitative analysis of MARCKS expression level and phosphorylation levels in HEK293 cells, hsSKM, and MEF cells at low and high densities. The data were normalized using the band intensity from HEK293 cells at low densities. n = 4. Statistical analysis was conducted using one-way ANOVA. *P < 0.05; **P < 0.01. **(B)** Cytoplasmic forced expression of HINT1 reduces MARCKS phosphorylation. HEK293A cells were seeded at low density, transfected with either WT HINT1-HA or mutant R24A/K25A HINT1-HA, incubated for 24 h, and subsequently lysed in SDS sample buffer. Phospho-MARCKS, total MARCKS, and actin levels were analyzed via Western blotting. Quantitative analysis of MARCKS phosphorylation levels in HEK293 cells, with or without transfection of WT HINT1-HA or mutant R24A/K25A HINT1-HA (right panel). The data were normalized using the band intensity from nontransfected cells. n = 3. Statistical analysis was conducted using a paired *t* test. *P < 0.05. **(C)** MARCKS phosphorylation level increases in HINT1 KO cells. WT and HINT1 KO cells were cultured overnight at low density, and subsequently lysed in SDS sample buffer. Phospho-MARCKS, total MARCKS, and actin levels were analyzed via Western blotting. Quantitative analysis of MARCKS phosphorylation levels in WT and HINT1 KO cells (right panel). The data were normalized using the band intensity from WT cells. n = 3. Statistical analysis was conducted using a paired *t* test. *P < 0.05. **(D)** Proposed model illustrates how HINT1 translocation is regulated by cell density and how this process contributes to the disruption of actin SFs, leading to a rounded, confined cell morphology. At low density, HINT1 predominantly localizes in the nucleus, where it binds to chromatin to regulate gene expression, thereby promoting cell proliferation and development. Cell–cell contact alone is insufficient to induce HINT1 translocation to the cytoplasm. At confluency, cells initiate the export of HINT1 to the cytoplasm, a process mediated by exportin 1. As cell density increases, more HINT1 is actively transported to the cytoplasm. In the cytoplasm, HINT1 inhibits phosphorylation of MARCKS, which influences actin remodeling, resulting in a rounded and compact cell morphology. This translocation also suppresses gene expression, leading to a halt in proliferation and promoting contact inhibition of proliferation. The subcellular localization of HINT1 appears to be cell type–dependent, which may explain why some cells form SFs, whereas others do not in situ. Source data are available for this figure.

# Discussion

We recently developed a DSP-MNase-proteogenomics method to identify chromatin-binding proteins that respond to mechanical stress and cell density (Li et al, 2023; Nakamura, 2024). Among the proteins identified was HINT1 from subconfluent (~80% confluency) hsSKM cells, although its role in mechanotransduction had not been previously explored. Although most proteins known to shuttle between the nucleus and cytoplasm in a cell density–dependent manner are also sensitive to mechanical force and substrate stiffness (Nakamura, 2024), we found that HINT1 is uniquely responsive to cell density but not to mechanical force.

HINT1 was originally identified as a tumor suppressor and an inhibitor of PKC (Robinson & Aitken, 1994). HINT1 is a member of the histidine triad (HIT) family, proteins that are recognized for their roles in nucleotide binding and hydrolysis (Chou et al, 2007). HINT1 also acts as a transcriptional suppressor via direct binding to transcription factors (Razin et al, 1999; Weiske & Huber, 2005; Wang et al, 2009).

HINT1 was first noted for its involvement in regulating cell growth and signal transduction and has since been linked to several key cellular functions, including apoptosis, gene expression regulation, and cellular stress response mechanisms (Dillenburg et al, 2023). Although HINT1 KO mice exhibit normal fetal and adult development, they show increased susceptibility to tumor formation and display increased anxiety-like behavior along with impaired motor coordination (Su et al, 2003; Li et al, 2006; Seburn et al, 2014; Sun et al, 2017).

HA-tagged HINT1 was initially reported to localize primarily in the cytoplasm of a human fibroblast cell line, with minimal presence in the nucleus (Brzoska et al, 1996). However, a subsequent study found that endogenous HINT1 predominantly localizes in the nucleus (Klein et al, 1998). Our findings may provide an explanation for this discrepancy, as transfected cells often reach high densities, even though only the transfected cells are visible via fluorescent probes. In addition, although HINT1 was primarily detected in the nucleus in human breast tissue sections, suggesting it does not function as a PKC inhibitor (Klein et al, 1998), this nuclear localization appears to occur in actively growing epithelial cells but not in deeper quiescent cells. Similarly, actively growing HeLa cells showed stronger nuclear HINT1 and less high-density cytoplasmic translocation versus nontransformed cells (HEK293A, hsSKM, MEF). This suggests a potential link between HINT1's nuclear role and tumorigenesis, though further validation with diverse cell types is required.

Given its small size (126–amino acid residues), it was surprising to find that HINT1 is not a mechanosensitive nucleocytoplasmic shuttling protein that can be imported through the extended nuclear pore complex (NPC). Although HINT1 forms dimers (Lima et al, 1996), its overall molecular weight still appears small enough to pass through NPC. According to the NPC stretching model (Elosegui-Artola et al, 2017), small molecules could be passively transported through a mechanically stretched NPC. For instance, we recently identified ubiquitin-conjugating enzyme E2 A (152 residues) and core-binding factor subunit beta (182 residues) as mechanosensitive and cell density–sensitive nucleocytoplasmic shuttling proteins (Feng et al, 2023a; Li et al, 2023; Li et al, 2024; Feng & Nakamura, 2025). In contrast, HINT1 translocation appears to be actively regulated by a transporter (Jans et al, 2019). Pharmacological experiments showed that HINT1's nuclear export is governed by exportin 1/CRM1, which is activated by high cell density but not by cell–cell contact, through an unknown pathway. Although the importin responsible for HINT1's nuclear import remains unidentified, importin $\alpha/\beta1$ has been ruled out based on pharmacological testing. A point mutation in the predicted intra-molecular eNLS impaired HINT1's nuclear entry at low density, indicating that an unknown importin must be involved in this process.

We also discovered that at high cell density, HINT1 translocation to the cytoplasm triggers the collapse of actin SFs, leading to a morphological shift in the cells, resulting in a smaller, rounder shape. This is also confirmed by the forced expression of HINT1 in the nucleus at low density, whereas normal HINT1 is mainly localized in the nucleus. Retaining HINT1 in the nucleus may not only promote its function for gene expression but also promote PKC activity in the cytoplasm, although we did not directly measure the latter. Active PKC in the cytoplasm may induce actin SFs through multiple pathways because many actin-binding proteins and signaling molecules involved in actin remodeling are PKC substrates (Larsson, 2006; Ringvold & Khalil, 2017). For example, in addition to the phosphorylation-dependent remodeling of actin filaments mediated by MARCKS, the PKC-AKT-Girdin pathway may also promote SF formation, as Girdin phosphorylation enhances SF assembly (Enomoto et al, 2005). Furthermore, HINT1 knockdown has been shown to increase PKC activity, as well as elevate both the expression and phosphorylation levels of AKT and Girdin (Wu et al, 2016; Garzon-Nino et al, 2017).

Translocation of HINT1 to the cytoplasm at high density may inhibit the PKC-mediated SF formation because HINT1 binds to a full-length PKC and inhibits its kinase activity (Klein et al, 1998; Zhang et al, 2021). Interestingly, reduction of HINT1 expression is observed in cardiac hypertrophic patients and the cross-sectional area of cardiomyocytes from HINT1-KO mice was larger than that of control mice in response to hypertrophic stimuli (Zhang et al, 2021), which is consistent with our observation in HINT1 KO cells.

In conclusion, our findings suggest that nucleocytoplasmic translocation of HINT1 regulates the disassembly of SF, through the dephosphorylation of MARCKS (Fig 7D). At low cell density, HINT1 remains in the nucleus, where it does not inhibit phosphorylation of MARCKS in the cytoplasm and binds to chromatin to regulate gene expression. Although cell–cell contact alone is insufficient to trigger HINT1's translocation, near-confluency induces its movement to the cytoplasm. As cell density increases, more HINT1 translocates to the cytoplasm, leading to a reduction in MARCKS phosphorylation at Ser167/170. This decreased phosphorylation may remodel actin filaments, resulting in a more rounded and compact cell shape that supports the maturation of tissue formation. Simultaneously, the removal of HINT1 from the nucleus suppresses gene expression, thereby promoting CIP. These findings are consistent with previous observations that CIP in Madin-Darby canine kidney cells proceeds in three distinct phases (Puliafito et al, 2012), yet our results further suggest that the final phase involves HINT1-driven actin remodeling at high cell density, independent of mechanical constraints.

Our work paves the way for future investigations. Key questions include whether HINT1 truly inhibits PKC specifically in the cytoplasm, and if so, whether it exhibits isoform specificity. Further research is also needed to explore the role of cell density in activating exportin and importin. In addition, our findings prompt investigations into the distinctions between cell–cell contact, confluency, and high density regarding intracellular structure and biological function, the factors driving these transitions, and how HINT1 loss prolongs tissue culture cell survival under high-density conditions.

# Materials and Methods

### Antibodies and reagents

Rabbit monoclonal anti-HINT1 antibody was purchased from Abcam (AB124912). Mouse monoclonal anti-YAP/TAZ antibody was purchased from Santa Cruz Biotechnology (sc-101199). Rabbit polyclonal anti-MARCKS (10004-2-Ig), phospho(Ser167/170)-MARCKS (29145-1-AP), and mouse monoclonal anti-$\beta$-catenin antibody (66379-1-Ig) were purchased from Proteintech. Anti-rabbit IgG Alexa Fluor Plus 594 antibody (A32754), anti-rabbit IgG Alexa Fluor 488 antibody (A11034), anti-mouse IgG Alexa Fluor Plus 594 antibody (A32744), phalloidin Alexa Fluor 488 (A12379), and phalloidin Alexa Fluor 568 (A12380) were purchased from Invitrogen. Mouse monoclonal anti-$\beta$-actin antibody (HC201-01), goat anti-rabbit IgG HRP antibody (HS101-01), and goat anti-mouse IgG HRP antibody (HS201-01) were purchased from TransGen. Hoechst 33342 was purchased from Thermo Fisher Scientific. CellBrite Fix 488 membrane stain was purchased from Biotium (30090-T). Latrunculin B was purchased from Sigma-Aldrich (428020). Blebbistatin was purchased from AbMole BioScience (856925-71-8). Leptomycin B was purchased from Solarbio (87081-35-4). Importazole (HY-101091) and ivermectin (HY-15310) were purchased from MedChemExpress.

### Plasmid construction

Primers used in this study are listed in Table S2. Human HINT1 cDNA (UniProt Accession ID: P49773) was amplified by PCR using 5′ primer, 5′-CCCAAGCTTATGGCAGATGAGATTGCCAAGGC-3′, and 3′ primer, 5′-CCGGAATTCACCAGGAGGCCAATGCATTTG-3′, using HEK293A cDNA library as a template and ligated into pcDNA3-HA vector at HindIII/EcoRI sites (Feng et al, 2023b).

aNLS-HINT1 and mutant aNLS-HINT1 were constructed by PCR using 5′ primer, 5′-CCCAAGCTTATGCCAAAAAAGAAGAGAAAGGTAATGGCAGATGAGATTGCCAAGGC-3′, 3′ primer, 5′-CCGGAATTCTTAACCAGGAGGCCAATGCATTTG-3′, and 5′ primer, 5′-CCCAAGCTTATGCCAAAAACGAAGAGAAAGGTAATGGCAGATGAGATTGCCAAGGC-3′, 3′ primer, 5′-CCGGAATTCTTAACCAGGAGGCCAATGCATTTG-3′, respectively, and ligated into pcDNA3-HA vector at HindIII/EcoRI sites.

Similarly, R24A/K25A mutant HINT1-HA was constructed using 5′ primer, 5′-CGATCTTTGGGAAGATCATCGCGGCGGAAATACCAGCCAAAATCATTTTTG-3′, and 3′ primer, 5′-GATGATCTTCCCAAAGATCG-3′, and

ligated into pcDNA3-HA by the T5 exonuclease DNA assembly (TEDA) method (Xia et al, 2019).

### Cell culture and transfection

HEK293A cells were purchased from Thermo Fisher Scientific. Human skeletal muscle (hsSKM) and MEF cells were purchased from the ATCC. These cells were grown in DMEM (Biological Industries) supplemented with 10% FBS (Biological Industries) and 1% penicillin–streptomycin. Cells were maintained at 37°C and 5% $CO_2$. Cells were transfected with polyethylenimine (PEI) or LipoGene2000 Star transfection reagent (L7002; US Everbright) in accordance with the manufacturer's protocol. To examine the effect of cell density on HINT1 localization, our preliminary experiments showed that seeding at $0.7 \times 10^5$ cells/$cm^2$ led to nearly complete cell-to-cell contact within 24 h, which we defined as 100% confluency (Fig S1). For reference, a seeding density of 200% corresponds to twice the number of cells required to achieve this confluency. However, because not all seeded cells adhere to the culture surface, higher values such as 400% do not reflect actual confluency but instead indicate the relative number of cells initially plated. Unless otherwise noted, we used 20% and 400% seeding densities to represent low and high cell density conditions, respectively.

### Cell culture on soft and stiff substrate

Cells were plated on the conventional cell culture six-well plates or CytoSoft six-well plates with soft (0.2 kPa) and stiff (64 kPa) substrates (#5165-5EA 0.2 kPa, #5145-5EA 64 kPa; Advanced Bio-Matrix) coated with 40 μg/ml fibronectin (03-090-1-01; Biological Industries). These cells were grown in DMEM (Biological Industries) supplemented with 10% FBS (Biological Industries) and 1% penicillin–streptomycin. Cells were maintained at 37°C and 5% $CO_2$.

### Immunofluorescence microscopy

Cells were plated on a 48-well plate coated with or without 10 ng/ml fibronectin (03-090-1-01, Biological Industries), washed once with PBS-D (PBS containing 1 mM $Ca^{2+}$ and $Mg^{2+}$), fixed with 4% formaldehyde in PBS-D for 20 min, rinsed in PBS-D, permeabilized with 0.5% Triton X-100 in TBS (50 mM Tris–HCl, pH 7.4, 150 mM NaCl) for 10 min, then rinsed again in TBS–0.1%Tx (TBS containing 0.1% Triton X-100), and blocked with 2% BSA in TBS–0.1%Tx for 1 h. Cells were then incubated with primary antibodies for 2 h, followed by several washes with TBS-0.1%Tx, incubated with secondary antibodies, and washed several more times with TBS–0.1%Tx. Cells were further incubated with phalloidin Alexa Fluor 488 or phalloidin Alexa Fluor 568, washed multiple times with TBS–0.1%Tx, and stained with Hoechst 33342 for 15 min. After a final wash, cell imaging was performed using EVOS FL Auto Imaging System (Thermo Fisher Scientific) with an EVOS Obj, Inf Plan Fluor 20 LWD objective, and Plan Fluor 40 LWD, .65NA/2.8WD objectives. Image processing and analysis were conducted using NIH ImageJ software (version 1.54 g).

## Immunofluorescence confocal microscopy

WT and HINT1-KO HEK293A cells were cultured on cover glasses coated with either polylysine or fibronectin. The cells were stained with CellBrite Fix 488 membrane stain for 15 min at 37°C, then fixed as described above, and prepared for imaging by being mounted with mounting medium (Spring Bioscience). Imaging was performed using a Leica SP8 confocal microscope equipped with a 63X HC PL APO objective lens with a numerical aperture (NA) of 1.40. The acquired images were processed and analyzed using ImageJ software.

## Western blotting

Cells were lysed in SDS or LDS sample buffer and heated at 95°C for 5 min. The samples were loaded onto either a homemade SDS–PAGE gel or a Novex NuPAGE 4–12% gradient Bis-Tris gel. Separated proteins were transferred to a nitrocellulose membrane and blocked with blocking buffer (5% nonfat milk powder in TBST: 20 mM Tris–HCl, pH 7.4, 110 mM NaCl, 5 mM MgCl$_2$, 0.1% Tween-20) for 2 h. Primary antibodies were prepared in this blocking solution, and the membranes were incubated overnight at 4°C. After washing in TBST, the membranes were incubated with HRP-conjugated secondary antibodies in blocking solution for 1 h at RT. The membranes were then washed and developed using HRP substrate (WesternBright ECL; Advansta).

## Knockout of HINT1 in tissue culture cells

HINT1 KO HEK293A cells were generated by delivery of Cas9 and target-specific guide RNAs (gRNAs). Oligos encoding the gRNAs for HINT1 were designed using CRISPick (https://portals.broadinstitute.org/gpp/public/analysis-tools/sgrna-design), and the selected HINT1-specific gRNA sequences, 5'-CACCGCCTGGTGGCGACACGATCTT-3' and 5'-AAACAAGATCGTGTCGCCACCAGGC-3', were cloned into BbsI-digested pX330-U6-Chimeric_BB-CBh-hSpCas9 (plasmid ID: 42,230; Addgene). Px330-sgHINT1-exon1 plasmids were transfected into HEK293A cells using LipoGene2000 Star transfection reagent (US Everbright) according to manufacturer's protocol. Briefly, HEK293A cells were seeded into a 24-well plate. After 24 h that the cells reached 60–70% confluency, 0.8 μg Px330-sgHINT1-exon1 plasmid was added to the well in the presence of LipoGene2000 Star transfection reagent. 72 h post-transfection, cells were then separated as single cells into a 96-well plate by serial dilution for another 7 d. Individual clones were expanded, and HINT1 protein expression was examined by immunoblotting.

## RNA sequencing

WT and HINT1 KO HEK293A cells were seeded at 5 × 10$^5$ cells in 60-mm dishes and cultured in DMEM supplemented with 10% FBS and 1% Pen/Strep for 18–24 h. Total RNA was isolated using the RNAiso Plus (Takara) and sent to Beijing Biomarker Technologies Co., Ltd. for library construction and sequencing. Briefly, extracted high-quality RNAs were used as input material for the RNA sample preparations. Sequencing libraries were generated according to

the following steps. Firstly, mRNA was purified from total RNA using poly-T oligo-attached magnetic beads. Fragmentation was carried out using divalent cations under elevated temperature in a fragmentation buffer. First strand cDNA was synthesized using random oligonucleotides. Second strand cDNA synthesis was subsequently performed using DNA polymerase I and RNase H. Remaining overhangs were converted into blunt ends via exonuclease/polymerase activities, and the enzymes were removed. After adenylation of the 30 ends of the DNA fragments, adapter oligonucleotides were ligated to prepare for hybridization. After post-ligation purification, DNA fragments with ligated adaptor molecules on both ends were selectively enriched in a 15-cycle PCR. Then, after post-amplification purification and quality control, the sequencing library was sequenced on the Illumina NovaSeq 6000 platform by Beijing BioMarker Technologies Co. Ltd.

## Cell migration assay

For random cell migration, WT and HINT1 KO HEK293A cells were seeded at 1 × 10$^5$ cells per well on a six-well plate. After 24 h, images were acquired for 20 h at 1 frame/10 min at 37°C using a ×10 Plan FL objective on an EVOS FL Auto time-lapse microscope with a monochrome and color camera. Cells were tracked using ImageJ (Plugin: Manual tracking) to obtain migration speed (μm/min). Cells that died, divided, or moved out of the frame were excluded from the analysis and tracking. The path of each cell was obtained as a track using ImageJ (Plugin: Chemotaxis tool).

## Correlation coefficient of HINT1 staining and Hoechst nuclear staining

To analyze the translocation of HINT1, the intensity of fluorescence along a predetermined line in an immunofluorescence image, which shows the nucleus in blue and the protein of interest in red, was measured using the "Plot Profile" function in ImageJ. To assess the relationship between the intensity levels in the blue (nucleus) and red (protein of interest) channels, and thereby gain insights into the localization of the protein within the cell, Pearson's correlation coefficient was calculated. This coefficient ranges from −1, indicating a complete cytosolic distribution, to 1, which signifies an exclusive nuclear presence, using GraphPad Prism 10.1.1.

## Quantification of cell cross-sectional area

We stained cell plasma membranes with CellBrite Fix 488 fluorogenic membrane dye (Biotium) following the manufacturer's protocol. We found that cells could not be fixed with formaldehyde before adding the dye because this caused the dye to penetrate the plasma membrane and stain intracellular membranes, blurring cell boundaries. We also observed that the dye itself caused cell shrinking at low cell densities. To analyze the cell cross-sectional area, the "segmentation" function of the Cellpose3 software (using the cyto 3 model, https://www.cellpose.org) was employed to detect cell boundaries in immunofluorescence

images and to generate mask files. These mask files were then imported into NIH ImageJ software (version 1.54 g), where the "Wand tool" and "ROI manager" functions were used to identify and record the selected regions. After a careful comparison with the original images, the accurately recorded selections were retained. Finally, a random sample of 100 cells from these recordings was selected for area calculations using the "Measure" function in ImageJ.

### Data analysis

For RNA-seq analysis on Galaxy (https://usegalaxy.org), raw reads were trimmed using TrimGalore (Galaxy Version 0.6.7 + galaxy0) and transcript abundance was quantified using Salmon quant (Galaxy Version 1.10.1 + galaxy2). Summarized transcript-level data were performed using tximport function (Galaxy Version 1.22.0). Differential expression and enrichment analyses were performed on iDEP (http://bioinformatics.sdstate.edu/idep/). Visualization and enrichment analysis of protein–protein interaction (PPI) network were performed using stringApp in Cytoscape software (version 3.10.3) (Shannon et al, 2003).

### Statistics

Data are the mean ± S.D. All experiments were performed at least three times independently. All image analysis was performed by operators who were blinded to the treatments administered. Significance was analyzed by $t$ test, one-way ANOVA followed by Tukey's multiple comparison test, two-way ANOVA followed by Tukey's multiple comparison test, or three-way ANOVA followed by Tukey's multiple comparison test. *$P \leqq 0.05$, **$P \leqq 0.01$, ***$P \leqq 0.001$, ****$P \leqq 0.0001$, ns: not significant. Statistical analysis was performed in GraphPad Prism 10.1.1.

## Data and Material Availability

Data that support the findings of this study are available within the article and its Supplementary figures and tables. The RNA-seq data generated from this study have been deposited in the Gene Expression Omnibus (GSE270956). All other data supporting the findings of this study are available from the corresponding author on reasonable request. Materials may be requested from the corresponding author.

## Supplementary Information

## Acknowledgements

We thank Tianjin University SPST Core Facility for providing support. This work was supported by the National Natural Science Foundation of China (32070777 to F Nakamura), the Tianjin University Graduate Education Grant (C1–2021–003), and the State Key Laboratory of Neurology and Oncology Drug Development (SKLSIM-F-202464).

### Author Contributions

X Zhang: data curation, formal analysis, validation, investigation, visualization, and writing—original draft.
F Nakamura: conceptualization, resources, supervision, funding acquisition, methodology, project administration, and writing—review and editing.

### Conflict of Interest Statement

The authors declare that they have no conflict of interest.

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
