## [Reviewer comments · Life Science Alliance]

Life Science Alliance

The nucleocytoplasmic translocation of HINT1 regulates the maturation of cell density

Xiaofang Zhang and Fumihiko Nakamura

DOI: <https://doi.org/10.26508/lsa.202503215>

Corresponding author(s): Fumihiko Nakamura, Tianjin University

Review Timeline:

Submission Date:	2025-01-13
Editorial Decision:	2025-03-14
Revision Received:	2025-05-01
Editorial Decision:	2025-06-16
Revision Received:	2025-07-09
Accepted:	2025-07-14

Scientific Editor: Tim Fessenden

Transaction Report:

March 14, 2025

Re: Life Science Alliance manuscript #LSA-2025-03215-T

Prof. Fumihiko Nakamura
Tianjin University
SPST
92 Weijin Road, Nankai District
Tianjin 300072
China

Dear Dr. Nakamura,

Thank you for submitting your manuscript entitled "The nucleocytoplasmic translocation of HINT1 regulates the maturation of cell density" to Life Science Alliance. The manuscript was assessed by expert reviewers, whose comments are appended to this letter. We invite you to submit a revised manuscript addressing the Reviewer comments.

Thank you for this interesting contribution to Life Science Alliance. We are looking forward to receiving your revised manuscript.

Sincerely,

B. MANUSCRIPT ORGANIZATION AND FORMATTING:

Reviewer #1 (Comments to the Authors (Required)):

Zhang et al. use three cell types (2 cell lines HEK293A, MEF; 1 primary cell type hsSKM) to study the shuttling of HINT1 from nucleus to cytoplasm in the context of cell monolayer density. The authors observed that HINT1 resides in the cell nucleus at low confluency, while it translocates to the cytoplasm at with increasing cell density. Alteration of substrate stiffness or actin function does not alter HINT1 localisation. HEK293A knockout cells for HINT1 did not show differences in cell shape, motility and proliferation, although they survive longer than control cells. They further analyze cross-sectional area as a measure for cell size. Pharmacological inhibition of exportin 1/CRM1 lead to retained HINT1 nuclear localization at high cellular density. Lastly, the authors investigate the phosphorylation of MARKS, a downstream regulator of PKC. Cells at high density exhibit decreased phosphorylation, similar to the effect observed with cytoplasmic forced expression of HINT1.

While the manuscript makes a solid effort to provide a detailed analysis of HINT1 localization, translocation, and pathway involvement, significant portions of the manuscript remain unclear, leading to conclusions that are not fully aligned with the presented data.

Major points:

1. Throughout the manuscript, the authors use the term "cells" to generalize their findings. However, cell type-specific differences are highly likely. The authors should refer for each statement to the specific cell types (or, in most cases, cell lines) they use in their study. Additionally, the manuscript presents selected cell types/lines inconsistently, making it difficult to follow the rationale for switching between them. To clarify, the authors should either consistently present the same cell type/line throughout the study or display all cell types/lines in parallel for comparison.
2. The concept of percentage density is confusing throughout the manuscript. Standardly, 100% confluency represents a fully confluent state, yet the authors describe this as only a near-full confluency, introducing additional confluency values ranging from 200% to 600%. Clarification is needed on how these values are calculated: do they correspond to the days in culture of the cells or to amount of seeded cells? Furthermore, the authors should specify what constitutes low and high density at seeding. Images in Figure S1 also show that 100% is not confluent, which needs further explanation.
3. The methodology for analyzing cross-sectional area needs to be clarified. Specifically, how can cells be larger at 600% confluency when no additional space is available for expansion while the cell numbers remain constant?
4. The authors use three HINT1-KO lines. It is unclear which clone was used in subsequent experiments. This should be explicitly stated to ensure reproducibility and clarity.
5. In Figure 2, cells at low confluency were treated with latrunculin B or blebbistatin, yet no change in the supposed nuclear localization of HINT1 was observed. Would treating high-density cells with these same small molecules alter HINT1 localization? If so, this could provide further insight into the mechanisms involved.
6. The authors expressed WT and NLS-mutated HINT1 and observed a change in cell shape in the latter case. Have they considered the possibility of a toxicity effect from the transfected construct? Additionally, the authors state that mutating an added NLS sequence does not affect HINT1 localization, but mutating the putative endogenous NLS results in HINT1 extrusion from the nucleus. How do the authors explain this discrepancy?
7. The authors claim to analyze PKC; however, their actual analysis pertains to the state of MARKS activation. This should be rephrased and specified to reflect the correct experimental approach and findings accurately.
8. Although the RNAseq enrichment analysis does not provide any insights, there are some molecules altered. Why did the authors decide not to follow up with functional analysis of highly differentially regulated transcripts?
9. Given that collagens are regulated and contribute to cell adhesion, could cell adhesion be rather compromised instead of intrinsic actin regulation? Actin regulation could be secondarily affected?

Minor Points:

1. Culture on stiffness gels is missing in the methods section.
2. Please spell out what the names YAP and CBF β stand for.
3. Fig. 3H: Why did the authors not show the pathways of cluster A?
4. Fig. 7A: It would be better to add the quantification of total and phosphorylated MARKS.
5. The first part of the abstract is really not very accurate and should be entirely changed. "Cell contact inhibition progresses through three stages: (i) increasing cell density leads to reduced movement while mitosis continues; (ii) a rapid shift to epithelial

morphology; and (iii) ongoing division with decreased cell size."

Reviewer #2 (Comments to the Authors (Required)):

This paper describes a dual functional protein, HINT1, which is a DNA binding protein that regulates gene expression in the nucleus and a PKC inhibitor that impacts cytoskeletal remodeling when in the nucleus. The authors show that HINT1 is found in the nucleus at low density and the cytoplasm at high density in 3 different cell types. Interestingly, unlike YAP, HINT1 is not influenced by actomyosin contractility or substrate stiffness. I have some suggestions to improve the manuscript, but overall think it is solid.

Major:

Are the cells in the monolayer 'mature' in the sense of producing adherens or tight junctions? Some staining might help reinforce the argument about the 'maturity' of the monolayer, at least in the HEK293A cells.

Regarding the RNAseq data shown in Fig. 3. Was the alteration in gene expression reproducible? It looks like there are some substantial differences between the replicates - as shown in the supplementary PCA.

It seems like it should be possible to squeeze a little more information out of this RNAseq dataset. Perhaps consult some bioinformatics colleagues on more refined GO or other analyses. Cytoscape might help you group these into informative clusters of interacting proteins. Some of the information in Fig. S4 could be expanded upon, and brought into the main text. Perhaps some of the genes that change the most or have suggestive functions (there are some interesting categories in the list!) could be tested for interaction (physical or regulatory) with HINT1?

Fig. 5C It is very difficult to see the actin structures. I see cortical actin but the stress fibers are not easy to see. Please include an inset or a higher magnification micrograph. It is also not surprising if a small, round, cell doesn't have stress fibers. Perhaps another experiment could be done in wild type cells in which a field of cells is stained for actin and there is a correlation (or not) between nuclear localization of HINT1 and stress fibers and/or cell spreading.

Fig. 6 clarify what is meant by 200% density and 300% density.

Fig. 7 assess MARCKS phosphorylation in HINT1 KO cells

Minor:

58. Specify which kinds of cells. Not all tissue culture cells behave in this manner.

86. Header : I think the Fig. 1 title makes a clearer subject header.

126. Change header to 'HINT1 localization is not responsive to mechanical cues'

138. I recommend a similar title for Fig. 2

148. Typo: 'Loss of HINT1 delays cell density...'

199 Typo: 'HINT1 KO enlarges cell cross-sectional area...'

209 Typos: 'To investigate the molecular mechanism of how translocation of HINT1 to the cytoplasm regulates cell morphology...'

255 Typo: two .. after 568

Reviewer #3 (Comments to the Authors (Required)):

This study addresses a fundamental yet often overlooked aspect of cell biology: how cells regulate their behavior in response to density. Basic scientific investigations like this are becoming increasingly rare, as the current research landscape tends to prioritize translational and applied studies. However, understanding these core mechanisms remains essential, as they provide the foundation for future biomedical advancements.

1. The authors have cited only one reference (reference number 3) to define the concept of cell contact inhibition. A broader discussion incorporating additional sources would strengthen the background and contextualization of this concept.

2. For low-density images, at least 20-30 cells should be presented separately (not necessarily as main figures but rather as supplementary data) to demonstrate that the observed localization is not limited to just one or two cells.

3. The study uses three cell lines: HUK293A, hsSKM, and MEF. However, it would be valuable to include a cancer cell line to examine how high cell density affects HINT1 translocation in a system where contact inhibition is overridden due to carcinogenesis.

4. Image analysis clarification: In Figure S1 (HINT1 translocation), the different densities (100%, 200%, 300%, and 400%) show no apparent differences upon visual inspection, especially in Figure S1. Could the authors clarify how the analysis was conducted and whether quantitative measures were used properly?

5. Figures 1A and 5A show that HINT1 expression in high-density cells appears marginal, with only two cells exhibiting HINT1 expression while the others remain negative. Could the authors explain and justify why HINT1 is sometimes expressed in all cells and, at other times, in only 1-2 cells, even under high-density culture conditions?
6. MARCKS phosphorylation is not exclusively mediated by PKC; it can also be phosphorylated by PKA and other kinases. To establish a direct link between HINT1 translocation and PKC activity, it may be beneficial to measure direct PKC activity in addition to MARCKS phosphorylation.

Life Science Alliance manuscript #LSA-2025-03215-T

Dr. Eric Sawey
Executive Editor
Life Science Alliance

Dear Dr. Eric Sawey,

Thank you for handling our manuscript (LSA-2025-03215-T) entitled “The nucleocytoplasmic translocation of HINT1 regulates the maturation of cell density”.

We also thank the reviewers for appreciating the value of the manuscript and providing very relevant comments that will help us to improve the manuscript. The revisions made in response to their feedback are indicated within the revised text using the “track changes” feature in MS Word and submitted as “Tracked Changes LSA-2025-03215-T”.

Point-by-point response to reviewers

Reviewer #1 (Comments to the Authors (Required)):

Zhang et al. use three cell types (2 cell lines HEK293A, MEF; 1 primary cell type hsSKM) to study the shuttling of HINT1 from nucleus to cytoplasm in the context of cell monolayer density. The authors observed that HINT1 resides in the cell nucleus at low confluency, while it translocates to the cytoplasm at with increasing cell density. Alteration of substrate stiffness or actin function does not alter HINT1 localisation. HEK293A knockout cells for HINT1 did not show differences in cell shape, motility and proliferation, although they survive longer than control cells. They further analyze cross-sectional area as a measure for cell size. Pharmacological inhibition of exportin 1/CRM1 lead to retained HINT1 nuclear localization at high cellular density. Lastly, the authors investigate the phosphorylation of MARKS, a downstream regulator of PKC. Cells at high density exhibit decreased phosphorylation, similar to the effect observed with cytoplasmic forced expression of HINT1.

While the manuscript makes a solid effort to provide a detailed analysis of HINT1 localization, translocation, and pathway involvement, significant portions of the manuscript remain unclear, leading to conclusions that are not fully aligned with the presented data.

We highly appreciate the reviewer’s careful reading and valuable suggestions which we incorporated to the manuscript as much as possible as follows. The changes made can be identified in the revised text through the "track changes" function in MS Words and submitted as “Tracked Changes LSA-2025-03215-T”. The line numbers refer to those in Tracked Changes LSA-2025-03215-T document.

Major points:

1. Throughout the manuscript, the authors use the term "cells" to generalize their findings. However, cell type-specific differences are highly likely. The authors should refer for each statement to the specific cell types (or, in most cases, cell lines) they use in their study. Additionally, the manuscript presents selected cell types/lines inconsistently, making it difficult to follow the rationale for switching between them. To clarify, the authors should either consistently present the same cell type/line throughout the study or display all cell types/lines in parallel for comparison.

Indeed, we observed cell type specificity. However, due to the difficulty of transfecting and generating knockout cell lines, we had to use different cell types in some cases. To clarify these limitations, we have included the rationale and the specific names of the cells used in the revised manuscript (line 124-128).

2. The concept of percentage density is confusing throughout the manuscript. Standardly, 100% confluency represents a fully confluent state, yet the authors describe this as only a near-full confluency, introducing additional confluency values ranging from 200% to 600%. Clarification is needed on how these values are calculated: do they correspond to the days in culture of the cells or to amount of seeded cells? Furthermore, the authors should specify

what constitutes low and high density at seeding. Images in Figure S1 also show that 100% is not confluent, which needs further explanation.

Thank you for pointing out this important detail that we had previously overlooked. Although many cell types typically reach around $1\sim 1.5\times 10^5$ cells/cm² at full confluency on tissue culture dishes, it's important to note that defining 100% confluency solely based on cell number is not entirely accurate, as it varies by cell type. Some cells can continue to grow and adopt a more confined shape even after initial contact, making the exact confluency difficult to determine. In our preliminary experiments, we found that seeding hsSKM cells at 0.7×10^5 cells/cm² generally led to 100% confluency after 24 hours. However, we could not precisely determine how much the cells would continue to crowd each other beyond initial contact, and some blank areas still remained after 24 hours of culture in some cases. Because achieving exact confluency is challenging, we seeded cells across a wide range of densities to ensure that we could clearly observe the effect of cell density on the subcellular localization of HINT1 and identify its overall trend. These explanations have been included in the revised manuscript (line 172-187) and Methods section (line 577-584).

3. The methodology for analyzing cross-sectional area needs to be clarified. Specifically, how can cells be larger at 600% confluency when no additional space is available for expansion while the cell numbers remain constant? An explanation of the cross-sectional area analysis has been included in the Methods section (line 683-691). The definition of confluency and its limitations are explained above.

4. The authors use three HINT1-KO lines. It is unclear which clone was used in subsequent experiments. This should be explicitly stated to ensure reproducibility and clarity. We apologize for the lack of clarity. Among the three HINT1-KO clones we generated, Clone KO-1-E9 was used for all subsequent experiments unless otherwise specified, as we did not observe any significant phenotypic differences among the KO lines. We have now explicitly stated this information in the revised manuscript (line 222-224).

5. In Figure 2, cells at low confluency were treated with latrunculin B or blebbistatin, yet no change in the supposed nuclear localization of HINT1 was observed. Would treating high-density cells with these same small molecules alter HINT1 localization? If so, this could provide further insight into the mechanisms involved. No, HINT1 localization at high cell density is not affected by treatment with either latrunculin B or blebbistatin. These new findings are presented in the supplementary data (Fig. S5, line 195-197).

6. The authors expressed WT and NLS-mutated HINT1 and observed a change in cell shape in the latter case. Have they considered the possibility of a toxicity effect from the transfected construct? Additionally, the authors state that mutating an added NLS sequence does not affect HINT1 localization, but mutating the putative endogenous NLS results in HINT1 extrusion from the nucleus. How do the authors explain this discrepancy? Yes, since transfection of the WT construct did not alter the cell phenotype, we believe the transfection procedure itself does not have a toxic effect. Regarding NLS, our results suggest that mutating the "added NLS" alone is insufficient to exclude HINT1 from the nucleus, likely due to the presence of an "endogenous NLS". To investigate this, we used an online prediction tool to identify a potential endogenous NLS within the HINT1 molecule. Mutation of this predicted endogenous NLS led to effective exclusion of HINT1 from the nucleus of the low density MEF cells. Therefore, we believe there is no discrepancy in our findings. To clarify this distinction, we refer to the "added NLS" as aNLS and the "endogenous NLS" as eNLS, respectively (line 296-308).

7. The authors claim to analyze PKC; however, their actual analysis pertains to the state of MARKS activation. This should be rephrased and specified to reflect the correct experimental approach and findings accurately. Due to limited resources and reagent availability, we assessed PKC activity by measuring the phosphorylation of MARCKS at S167/S170, which are currently known to be specifically targeted by PKC. We have clarified and detailed these points in the revised manuscript (line 376-382).

8. Although the RNAseq enrichment analysis does not provide any insights, there are some molecules altered. Why did the authors decide not to follow up with functional analysis of highly differentially regulated transcripts? Thank you for pointing out this interesting possibility. As it requires more in-depth and rigorous analysis, we plan to further investigate the highly differentially expressed transcripts in future studies. In the present study, however, we focused on the function of HINT1 translocation (line 232-238)

9. Given that collagens are regulated and contribute to cell adhesion, could cell adhesion be rather compromised instead of intrinsic actin regulation? Actin regulation could be secondarily affected?

Thank you for bringing up this possibility. Currently, we lack evidence to determine whether collagen directly contributes to the regulation of cell density. However, because various collagen genes are both up- and down-regulated in the knockout condition, we presume that collagen is unlikely to be a decisive factor in cell density regulation—particularly if these collagens share similar roles in cell adhesion. In light of the reviewer's valuable suggestion, we have addressed this point in the main text (line 238-242)

Minor Points:

1. Culture on stiffness gels is missing in the methods section.

Added (line 586-592).

2. Please spell out what the names YAP and CBF β stand for.

Defined (line 164 and 190-191).

3. Fig. 3H: Why did the authors not show the pathways of cluster A?

Cluster A displays higher variability within each group (WT and KO), whereas clusters B and D show distinct differences between WT and KO cells (line 264-265). Accordingly, we focused our analysis on clusters B and D. In response to Reviewer #2's request for a more rigorous RNA-seq analysis, we reanalyzed the dataset using Cytoscape. The new results are presented in the supplementary data (Fig. S7G) and were largely consistent with our original findings.

4. Fig. 7A: It would be better to add the quantification of total and phosphorylated MARKS.

Quantified results from multiple experiments are shown in Fig. 7A (line 384-386).

5. The first part of the abstract is really not very accurate and should be entirely changed. "Cell contact inhibition progresses through three stages: (i) increasing cell density leads to reduced movement while mitosis continues; (ii) a rapid shift to epithelial morphology; and (iii) ongoing division with decreased cell size."

We have revised the wording to improve accuracy.

Reviewer #2 (Comments to the Authors (Required)):

This paper describes a dual functional protein, HINT1, which is a DNA binding protein that regulates gene expression in the nucleus and a PKC inhibitor that impacts cytoskeletal remodeling when in the nucleus. The authors show that HINT1 is found in the nucleus at low density and the cytoplasm at high density in 3 different cell types. Interestingly, unlike YAP, HINT1 is not influenced by actomyosin contractility or substrate stiffness. I have some suggestions to improve the manuscript, but overall think it is solid.

We greatly appreciate the reviewer's valuable comments and suggestions, which we have incorporated into the revised manuscript as thoroughly as possible. The changes made can be identified in the revised text through the "track changes" function in MS Words and submitted as "Tracked Changes LSA-2025-03215-T". The line numbers refer to those in Tracked Changes LSA-2025-03215-T document.

Major:

Are the cells in the monolayer 'mature' in the sense of producing adherens or tight junctions? Some staining might help reinforce the argument about the 'maturity' of the monolayer, at least in the HEK293A cells.

β -Catenin staining was performed to assess cell adhesion formation, and the results are included in the supplementary data (Fig. S3, line 171-172).

Regarding the RNAseq data shown in Fig. 3. Was the alteration in gene expression reproducible? It looks like there are some substantial differences between the replicates - as shown in the supplementary PCA.

It seems like it should be possible to squeeze a little more information out of this RNAseq dataset. Perhaps consult some bioinformatics colleagues on more refined GO or other analyses. Cytoscape might help you group these into informative clusters of interacting proteins. Some of the information in Fig. S4 could be expanded upon, and

brought into the main text. Perhaps some of the genes that change the most or have suggestive functions (there are some interesting categories in the list!) could be tested for interaction (physical or regulatory) with HINT1?

Since this variance is commonly seen in the RNAseq data, we believe that the data represent the reproducible results. Since the current report focuses on the translocation of HINT1, we did not much delve into these differentially expressed genes in this paper. However, we agree that further analysis would be helpful to understand the insight of the function of HINT1. Therefore, we used Cytoscape to analyze our data and added in the supplementary data (Fig. S7G). However, the new result was largely consistent with our original findings (line 230-232).

Fig. 5C It is very difficult to see the actin structures. I see cortical actin but the stress fibers are not easy to see. Please include an inset or a higher magnification micrograph. It is also not surprising if a small, round, cell doesn't have stress fibers. Perhaps another experiment could be done in wild type cells in which a field of cells is stained for actin and there is a correlation (or not) between nuclear localization of HINT1 and stress fibers and/or cell spreading.

The enlarged image is presented as an inset in Fig. S11.

Fig. S2 shows correlation between nuclear localization of HINT1 and SF in cells.

Fig. 6 clarify what is meant by 200% density and 300% density.

Thank you for pointing out this important detail that we had previously overlooked. Although many cell types typically reach full confluency at around $1-1.5 \times 10^5$ cells/cm² on tissue culture dishes, 100% confluency cannot be strictly determined by cell number alone, as it varies depending on the cell type. Some cells continue to proliferate and adopt a more compact morphology after initial contact, making it difficult to establish a consistent threshold. In our preliminary experiments, we observed that seeding at 0.7×10^5 cells/cm² resulted in nearly all cells contacting one another after 24 hours, which we defined as 100% confluency. However, the extent of further crowding beyond initial contact was difficult to quantify, and in some cases, small gaps remained. Therefore, a seeding density of 200% refers to twice the number of cells required to reach 100% confluency. Since not all seeded cells attach, values like 300% do not represent actual confluency but rather the relative number of cells initially plated. These explanations have been included in the revised manuscript (line 172-187) and methods section (line 577-584).

Fig. 7 assess MARCKS phosphorylation in HINT1 KO cells

We compared the phosphorylation levels of MARCKS between WT and HINT1-KO cells, as shown in Fig. 7C. The results revealed higher phosphorylation in the KO cells compared to WT, consistent with our hypothesis.

Minor:

58. Specify which kinds of cells. Not all tissue culture cells behave in this manner.

Specified throughout the manuscript.

86. Header : I think the Fig. 1 title makes a clearer subject header.

Changed

126. Change header to 'HINT1 localization is not responsive to mechanical cues'

Changed.

138. I recommend a similar title for Fig. 2

Changed.

148. Typo: 'Loss of HINT1 delays cell density...'

Amended.

199 Typo: 'HINT1 KO enlarges cell cross-sectional area...'

Amended.

209 Typos: 'To investigate the molecular mechanism of how translocation of HINT1 to the cytoplasm regulates cell morphology...'

Amended.

255 Typo: two .. after 568

Amended.

Reviewer #3 (Comments to the Authors (Required)):

This study addresses a fundamental yet often overlooked aspect of cell biology: how cells regulate their behavior in response to density. Basic scientific investigations like this are becoming increasingly rare, as the current research landscape tends to prioritize translational and applied studies. However, understanding these core mechanisms remains essential, as they provide the foundation for future biomedical advancements.

We are grateful for the reviewer's positive feedback and valuable suggestions, which we have incorporated into the manuscript as much as possible. The changes made can be identified in the revised text through the "track changes" function in MS Words and submitted as "Tracked Changes LSA-2025-03215-T". The line numbers refer to those in Tracked Changes LSA-2025-03215-T document.

1. The authors have cited only one reference (reference number 3) to define the concept of cell contact inhibition. A broader discussion incorporating additional sources would strengthen the background and contextualization of this concept.

Two references were originally cited to define the concept of contact inhibition. We have now added a more recent reference following that sentence (line 63-64).

2. For low-density images, at least 20-30 cells should be presented separately (not necessarily as main figures but rather as supplementary data) to demonstrate that the observed localization is not limited to just one or two cells. Additional images have been included in manuscript (Fig. 1A) and Supplementary data (Fig. S1A).

3. The study uses three cell lines: HUK293A, hsSKM, and MEF. However, it would be valuable to include a cancer cell line to examine how high cell density affects HINT1 translocation in a system where contact inhibition is overridden due to carcinogenesis.

Thank you for this valuable suggestion. To explore this, we examined HINT1 localization in actively growing HeLa cells at both low and high densities. As the reviewer anticipated, the localization behavior of HINT1 in HeLa cells differed from that observed in HEK293A, hsSKM, and MEF cells. The results have been added to the supplementary data (Fig. S1B, C) and are described in the main text (lines 473–475).

4. Image analysis clarification: In Figure S1 (HINT1 translocation), the different densities (100%, 200%, 300%, and 400%) show no apparent differences upon visual inspection, especially in Figure S1. Could the authors clarify how the analysis was conducted and whether quantitative measures were used properly?

Although many cell types typically reach full confluency at around $1-1.5 \times 10^5$ cells/cm² on tissue culture dishes, 100% confluency cannot be strictly determined by cell number alone, as it varies depending on the cell type. Some cells continue to proliferate and adopt a more compact morphology after initial contact, making it difficult to establish a consistent threshold. In our preliminary experiments, we observed that seeding at 0.7×10^5 cells/cm² resulted in nearly all cells contacting one another after 24 hours, which we defined as 100% confluency. However, the extent of further crowding beyond initial contact was difficult to quantify, and in some cases, small gaps remained. Therefore, a seeding density of 200% refers to twice the number of cells required to reach 100% confluency. Since not all seeded cells attach, values like 400% do not represent actual confluency but rather the relative number of cells initially plated. To address this limitation and reliably assess the impact of cell density on HINT1 subcellular localization, we seeded cells across a broad range of densities. This approach allowed us to capture overall trends, even though no obvious differences were observed upon visual inspection. These clarifications have been added to the revised manuscript (line 172-187) and methods section (line 577-584).

To analyze HINT1 translocation, we measured fluorescence intensity along a defined line in immunofluorescent images, where the nucleus is stained blue and the protein of interest is labeled in red. This was performed using the "Plot Profile" function in ImageJ. To evaluate the spatial relationship between the nuclear (blue) and protein (red) signals—thereby assessing subcellular localization—we calculated the Pearson's correlation coefficient using GraphPad Prism 10.1.1. This coefficient ranges from -1, indicating complete cytoplasmic

localization, to 1, indicating exclusive nuclear localization. We believe this approach effectively reflects the nucleocytoplasmic translocation of HINT1. This explanation has been added to the Methods section (line 670-681).

5. Figures 1A and 5A show that HINT1 expression in high-density cells appears marginal, with only two cells exhibiting HINT1 expression while the others remain negative. Could the authors explain and justify why HINT1 is sometimes expressed in all cells and, at other times, in only 1-2 cells, even under high-density culture conditions? In these figures, only the transfected cells were stained with anti-HA-tag antibodies, which is why not all cells show staining.

6. MARCKS phosphorylation is not exclusively mediated by PKC; it can also be phosphorylated by PKA and other kinases. To establish a direct link between HINT1 translocation and PKC activity, it may be beneficial to measure direct PKC activity in addition to MARCKS phosphorylation.

Since the inhibition of PKC by HINT1 is well established, and Ser167 and Ser170 are the only known phosphorylation sites on MARCKS specifically targeted by PKC (whereas PKA phosphorylates different sites), we used a site-specific anti-phospho antibody to assess PKC activity. The antibody employed specifically recognizes phosphorylation at Ser167/170—sites currently known to be phosphorylated by PKC but not by other kinases, as documented in PhosphoSitePlus (<https://www.phosphosite.org/proteinAction.action?id=996&showAllSites=true>). To date, no other kinase has been reported to target these specific residues. Therefore, we evaluated PKC activity by detecting phosphorylation of MARCKS at these sites. However, as the reviewer correctly pointed out, the involvement of other kinases cannot be entirely ruled out. Due to limitations in our resources and reagent availability, we were unable to directly measure PKC activity. Nonetheless, we believe that phosphorylation of MARCKS at Ser167/170 serves as a reliable proxy for PKC activity. We have revised the conclusion in the manuscript to reflect this with greater accuracy.

June 16, 2025

RE: Life Science Alliance Manuscript #LSA-2025-03215-TR

Prof. Fumihiko Nakamura
Tianjin University
SPST
92 Weijin Road, Nankai District
Tianjin 300072
China

Dear Dr. Nakamura,

Thank you for submitting your revised manuscript entitled "The nucleocytoplasmic translocation of HINT1 regulates the maturation of cell density". This manuscript was returned to the original reviewers whose comments are below.

As you will see, reviewers overall appreciated the significant changes made to improve this work however Reviewers 1 and 3 noted some concerns on methodology and claims made in the text that are still outstanding. We concur that the details added to the text in response to Reviewer 1 should be moved to the methods section. Meanwhile the much-needed explanation of confluency calculations only come after data in Figure 1 has been presented. This text should be moved earlier to introduce the methodology used throughout. Finally, please amend claims throughout to refer to MARCKs and not PKC activity when the latter was not directly measured. Reviewer 3 noted that claims on HeLa cells were not substantiated in the data. We agree that the difference in HINT1 between confluent HeLa and confluent nontransformed cells is not formally tested so this claim should be removed. We would be happy to publish your paper in Life Science Alliance pending these changes and final revisions necessary to meet our formatting guidelines.

- Please be sure that the authorship listing and order is correct.
- Please upload your main and supplementary figures as single files.
- Please add the X and Bluesky handles of your host institute/organization, as well as your own and/or one of the authors, to our system.
- It is recommended to exclude figures from the manuscript text and upload them separately.
- After the references section, please add your main, supplementary figure, and table legends to the main manuscript text.
- Please add callouts for Figures 4A-B; S2A-B; S4A-C; S5A-B; S6A-B; S7A-G; S9A-C; S10A-B; S12A-B; S13 A-D and Table S2 to your main manuscript text.

LSA now encourages authors to provide a 30-60 second video where the study is briefly explained. We will use these videos on social media to promote the published paper and the presenting author (for examples, see <https://docs.google.com/document/d/1-UWCfbE4pGcDdcgzcmiuJI2XMBJnxKYeqRvLLrLS08s/edit?usp=sharing>). Corresponding or first-authors are welcome to submit the video. Please submit only one video per manuscript. The video can be emailed to contact@life-science-alliance.org

A. FINAL FILES:

- An editable version of the final text (.DOC or .DOCX) is needed for copyediting (no PDFs).
- High-resolution figure, supplementary figure and video files uploaded as individual files: See our detailed guidelines for preparing your production-ready images, <https://www.life-science-alliance.org/authors>
- Summary blurb (enter in submission system): A short text summarizing in a single sentence the study (max. 200 characters)

including spaces). This text is used in conjunction with the titles of papers, hence should be informative and complementary to the title. It should describe the context and significance of the findings for a general readership; it should be written in the present tense and refer to the work in the third person. Author names should not be mentioned.

B. MANUSCRIPT ORGANIZATION AND FORMATTING:

Sincerely,

Reviewer #1 (Comments to the Authors (Required)):

Zhang and Nakamura have made a commendable effort in addressing the points I previously raised. It can be acknowledged that the authors have added information to the results section to address my main concerns (Points 1 and 2). However, I believe this additional detail disrupts the flow of the results, and most of it would be more appropriately placed in the Methods section. A brief clarification in the results would suffice.

The explanation of the concept of confluency remains still unclear and would benefit from simplification.

Furthermore, the authors still make inaccurate claims, such as "inhibiting protein kinase C" (line 53), "an antibody to evaluate PKC activity" (line 380), and "leading to a reduction in PKC activity" (line 518). These statements are not factually correct and should be revised for greater accuracy and scientific rigor.

Reviewer #2 (Comments to the Authors (Required)):

The authors have addressed all of my concerns.

Reviewer #3 (Comments to the Authors (Required)):

I appreciate the authors' effort in revising the manuscript and responding to the reviewer feedback in detail. Most of my initial concerns have been adequately addressed. However, I would like to raise a few remaining points regarding the imaging data provided in response to comments 2 and 3:

While I appreciate the authors' efforts to address the previous comments, I still have some concerns regarding the imaging data. In response to comment 2, the high-density images show clear HINT1 translocation only in MEF cells, whereas in hsSKM cells, HINT1 remains predominantly nuclear, and the translocation is not as evident. Moreover, the immunofluorescence staining patterns in Figures S1 and S2 for hsSKM cells appear quite different, which raises questions about consistency. This limits the generalizability of the findings across non-cancerous cell types. Regarding comment 3, the images provided for HeLa cells do not clearly support the authors' claim that HINT1 is localized to both the nucleus and cytoplasm at high density; the resolution is insufficient, and clearer, higher-resolution images or additional quantification are needed to validate this observation.

Life Science Alliance manuscript #LSA-2025-03215-TR

Dr. Tim Fessenden
Life Science Alliance

Dear Dr. Fessenden,

We appreciate you processing our manuscript (LSA-2025-03215-TR) entitled "The nucleocytoplasmic translocation of HINT1 regulates the maturation of cell density" and are pleased to learn that the suggested minor revisions offer the potential for acceptance.

Our gratitude also extends to the reviewers for recognizing the manuscript's value and for their invaluable constructive feedback. The revisions addressing their comments are clearly marked in the accompanying " Tracked Changes LSA-2025-03215-TR " document, generated using Microsoft Word's feature.

Point-by-point response to reviewers

As you will see, reviewers overall appreciated the significant changes made to improve this work however Reviewers 1 and 3 noted some concerns on methodology and claims made in the text that are still outstanding. We concur that the details added to the text in response to Reviewer 1 should be moved to the methods section. Meanwhile the much-needed explanation of confluency calculations only come after data in Figure 1 has been presented. This text should be moved earlier to introduce the methodology used throughout. Finally, please amend claims throughout to refer to MARCKs and not PKC activity when the latter was not directly measured. Reviewer 3 noted that claims on HeLa cells were not substantiated in the data. We agree that the difference in HINT1 between confluent HeLa and confluent nontransformed cells is not formally tested so this claim should be removed. We would be happy to publish your paper in Life Science Alliance pending these changes and final revisions necessary to meet our formatting guidelines.

Thank you for organizing the remaining issues. We've addressed each point as follows:

First, the explanation of the confluency calculations has been moved earlier in the manuscript and is now presented in a new Fig. S1 (formerly Fig. S3). Consequently, the numbering of all subsequent supplementary figures has been adjusted.

Second, all statements referring to PKC activity have been revised for accuracy throughout the manuscript, as suggested, to reflect that direct measurements were not performed.

Finally, regarding Reviewer 3's comment, we have investigated and quantified the distribution of HINT1 at various confluency levels. We found a quantitative difference between transformed HeLa cells and non-transformed cells, which is now shown in a new Fig. S4. We acknowledge that findings from HeLa cells alone cannot be generalized, and this limitation is now clearly stated in the discussion.

-Please be sure that the authorship listing and order is correct.

Confirmed.

-Please upload your main and supplementary figures as single files.

Uploaded as suggested.

-Please add the X and Bluesky handles of your host institute/organization, as well as your own and/or one of the authors, to our system.

These SNSs are not accessible from China.

-It is recommended to exclude figures from the manuscript text and upload them separately.
The figures have been removed from the manuscript text and uploaded as individual files.

-After the references section, please add your main, supplementary figure, and table legends to the main manuscript text.
The legends for the main figures, supplementary figures, and tables have been relocated to the main manuscript text, following the references section, as requested.

-Please add callouts for Figures 4A-B; S2A-B; S4A-C; S5A-B; S6A-B; S7A-G; S9A-C; S10A-B; S12A-B; S13 A-D and Table S2 to your main manuscript text.
We have incorporated the requested callouts for the Figures.

We appreciate your clear guidance on the necessary revisions. Please find our response to the reviewers outlined below.

Reviewer #1 (Comments to the Authors (Required)):

Zhang and Nakamura have made a commendable effort in addressing the points I previously raised. It can be acknowledged that the authors have added information to the results section to address my main concerns (Points 1 and 2). However, I believe this additional detail disrupts the flow of the results, and most of it would be more appropriately placed in the Methods section. A brief clarification in the results would suffice.

The explanation of the concept of confluency remains still unclear and would benefit from simplification.

We appreciate the reviewer's valuable suggestion and have rearranged the flow accordingly (new line 125-142).

Furthermore, the authors still make inaccurate claims, such as "inhibiting protein kinase C" (line 53), "an antibody to evaluate PKC activity" (line 380), and "leading to a reduction in PKC activity" (line 518). These statements are not factually correct and should be revised for greater accuracy and scientific rigor.

All specified statements regarding PKC activity have been revised for accuracy throughout the manuscript, as suggested.

Reviewer #2 (Comments to the Authors (Required)):

The authors have addressed all of my concerns.

Thank you.

Reviewer #3 (Comments to the Authors (Required)):

I appreciate the authors' effort in revising the manuscript and responding to the reviewer feedback in detail. Most of my initial concerns have been adequately addressed. However, I would like to raise a few remaining points regarding the imaging data provided in response to comments 2 and 3:

While I appreciate the authors' efforts to address the previous comments, I still have some concerns regarding the imaging data. In response to comment 2, the high-density images show clear HINT1

translocation only in MEF cells, whereas in hsSKM cells, HINT1 remains predominantly nuclear, and the translocation is not as evident. Moreover, the immunofluorescence staining patterns in Figures S1 and S2 for hsSKM cells appear quite different, which raises questions about consistency. This limits the generalizability of the findings across non-cancerous cell types.

Thank you for your valuable effort to improve our manuscript.

Responding to the reviewer's previous comment 2 (For low-density images, at least 20-30 cells should be presented separately (not necessarily as main figures but rather as supplementary data) to demonstrate that the observed localization is not limited to just one or two cells.), we have provided lower magnification images to show a greater number of cells (Fig. S2). A technical issue led to these images also losing resolution. We suspect this reduced resolution might have given the impression that HINT1's cytoplasmic translocation in hsSKM cells is less clear than it is. Nevertheless, Fig. 1D (higher magnification) and the revised Fig. S2 (lower magnification) both clearly demonstrate the translocation.

Regarding comment 3, the images provided for HeLa cells do not clearly support the authors' claim that HINT1 is localized to both the nucleus and cytoplasm at high density; the resolution is insufficient, and clearer, higher-resolution images or additional quantification are needed to validate this observation.

We suspected the density-dependency of HINT1 translocation in transformed HeLa cells might differ from non-transformed cells. To investigate this, we plated HeLa cells at various densities and quantified HINT1 translocation (Fig S4). We found that actively growing HeLa cells showed stronger nuclear HINT1 and less cytoplasmic translocation at high densities compared to non-transformed cells (HEK293A, hsSKM, MEF). This suggests a potential link between HINT1's nuclear role and tumorigenesis, though further validation with diverse cell types is required. This point is also stated in the discussion.

July 14, 2025

RE: Life Science Alliance Manuscript #LSA-2025-03215-TRR

Prof. Fumihiko Nakamura
Tianjin University
SPST
92 Weijin Road, Nankai District
Tianjin 300072
China

Dear Dr. Nakamura,

Thank you for submitting your Research Article entitled "The nucleocytoplasmic translocation of HINT1 regulates the maturation of cell density". It is a pleasure to let you know that your manuscript is now accepted for publication in Life Science Alliance. Congratulations on this interesting work.

DISTRIBUTION OF MATERIALS:

Again, congratulations on a very nice paper. I hope you found the review process to be constructive and are pleased with how the manuscript was handled editorially. We look forward to future exciting submissions from your lab.

Sincerely,
